# Resolution limit of image analysis algorithms

Edward A.K. Cohen[1], Anish V. Abraham[2,3], Sreevidhya Ramakrishnan[2,3] & Raimund J. Ober[2,3,4]

The resolution of an imaging system is a key property that, despite many advances in optical imaging methods, remains difficult to define and apply. Rayleigh's and Abbe's resolution criteria were developed for observations with the human eye. However, modern imaging data is typically acquired on highly sensitive cameras and often requires complex image processing algorithms to analyze. Currently, no approaches are available for evaluating the resolving capability of such image processing algorithms that are now central to the analysis of imaging data, particularly location-based imaging data. Using methods of spatial statistics, we develop a novel algorithmic resolution limit to evaluate the resolving capabilities of location-based image processing algorithms. We show how insufficient algorithmic resolution can impact the outcome of location-based image analysis and present an approach to account for algorithmic resolution in the analysis of spatial location patterns.

[1] Department of Mathematics, Imperial College London, London SW7 2AZ, UK. [2] Department of Biomedical Engineering, Texas A&M University, College Station, TX 77843, USA. [3] Department of Molecular & Cellular Medicine, Texas A&M University, College Station, TX 77843, USA. [4] Centre for Cancer Immunology, Faculty of Medicine, University of Southampton, Southampton SO16 6YD, UK. Correspondence and requests for materials should be addressed to E.A.K.C. (email: e.cohen@imperial.ac.uk) or to R.J.O. (email: raimund.ober@tamu.edu)

Resolution is one of the most important properties of an imaging system, yet it remains difficult to define and apply. Rayleigh's and Abbe's resolution criteria[1] were developed for observations with the human eye and had a major influence on the development of optical instruments. However, no systematic approach is yet available for the evaluation of the often complex image processing algorithms that have become central to the analysis of the imaging data that today is acquired by highly sensitive cameras. This is particularly relevant for the many modern imaging experiments and corresponding image processing algorithms for which the detection of objects (e.g., molecules, molecular complexes, subcellular organelles) form an integral aspect. Examples are localization-based superresolution experiments (PALM, STORM, etc.[2–4]), experiments to investigate the arrangement of molecular complexes on the cellular membrane such as clathrin-coated pits[5,6], experiments tracking single particles[7,8] or subcellular organelles[9], etc.

Common to the analysis of experimental data produced by such "object-based" imaging experiments is the central role that image analysis algorithms play in the identification and localization of the underlying objects, be they single molecules, clathrin-coated pits, etc. The success of such imaging experiments is, therefore, to a large extent dependent on how well these algorithms can resolve the imaged objects[10]. The assessment of such algorithms in terms of their resolution capabilities is, however, largely unexplored.

Here, we use methods of spatial statistics to quantitatively evaluate the resolution capabilities of location-based image analysis algorithms and to demonstrate the impact of resolution limitations on the analysis of object-based imaging data. A specific example that we will consider in detail relates to the question of whether the distribution of clathrin-coated pits is purely random or exhibits other spatial characteristics such as clustering. Methods of spatial statistics, which have been extensively used in different scientific disciplines[11], form the theoretical background for the development of this manuscript and underpin the presented methods for evaluating location-based image analysis algorithms. This theoretical background is introduced in Supplementary Note 1 and rigorously developed in Supplementary Notes 2–7. Central to this analysis is the notion of algorithmic resolution which we introduce to characterize an algorithm's ability to resolve objects.

## Results

### Detecting the effect of algorithmic resolution

First, we show that insufficient "algorithmic resolution" of an image analysis algorithm can have a significant impact on the outcome of the analysis of spatial patterns which is typically carried out using the pair-correlation function or Ripley's $K$-function[11] (see Supplementary Note 2). For a spatial pattern that is uniformly distributed (in the probabilistic sense), also termed completely spatially random, the pair-correlation function $g$, which describes the relationship between pairs of objects that are a distance $r$ apart, is given by the identity function $g(r) = 1$, $r > 0$. Ripley's $K$-function, which describes the expected number of objects within a distance $r$ of an arbitrary object, is given by $K(r) := \pi r^2$, $r > 0$, for a completely spatially random pattern. This implies that the related $L(r) - r$ function, where $L(r) := \sqrt{(K(r)/\pi)}$ for $r > 0$, is constant and equal to zero. The $L(r) - r$ function is non-zero if the point pattern is not completely spatially random. Clustering, in which objects are typically closer to each other than one would expect under complete spatial randomness, is characterized through positive values of this function, whereas deviations from 0 to negative values indicates inhibition or regularity, meaning that the spacing of points is somewhat larger than that in completely

spatially random data. Here, it is also instructive to recall that completely spatially random data are those in which the events occur completely at random and independently of each other. So at first sight some spatial configurations of events might be seen that resemble clusters, whereas in other areas large "empty" patches might be seen (see Supplementary Figure 1). These occur purely by chance and are not due to some underlying correlation structure within the data. However, importantly for our considerations, all possible spatial configurations of events are sampled.

An important question in cell biology is whether or not structures are organized in a regular way or do not have particular relations among them. Clathrin-coated pits play a major role in endocytosis. Whether clathrin-coated pits are positioned in an ordered fashion is of major interest in cell biology. Translated into the language of spatial statistics, we are therefore interested in whether or not clathrin-coated pits are distributed in a completely spatially random fashion[5,6]. This question itself can be addressed by investigating the $L(r) - r$ function of the locations of the pits.

The clathrin-coated pit imaging data of Fig. 1a was processed using several established algorithms (see the list of image analysis approaches in Methods) to determine the locations of the pits (Fig. 1b, c), which were then further analyzed by plotting the estimated $\hat{L}(r) - r$ function (Fig. 1d). The analysis appears to show that the pits are not distributed in a completely spatially random fashion as the $\hat{L}(r) - r$ plot is not equal to zero for all the processing schemes, thereby suggesting a nonuniform arrangement of the pits on the plasma membrane. To understand this behavior, we simulated clathrin-coated pits that are located according to a completely spatially random distribution (Fig. 1e). Estimating and analyzing the locations of these simulated pits (Fig. 1f, g) in the same fashion as done for the experimentally acquired data reveals that the resulting $\hat{L}(r) - r$ plots show remarkable similarity with those obtained from the experimentally acquired data (Fig. 1h). Importantly, these plots do not show a constant value of 0 as would be expected for completely spatially random data. This suggests that the deviations from the expected constant appearance of the $\hat{L}(r) - r$ function are due to effects of the data analysis rather than being a property of the distribution of the clathrin-coated pits.

To further understand this phenomenon, we investigated whether the observed effects might be due to the different capabilities of the image processing algorithms to resolve clathrin-coated pits. To do this, in Supplementary Notes 3–5 we theoretically analyzed the impact of limited "algorithmic resolution" of an image analysis algorithm on the pair-correlation and the $L(r) - r$ functions. We modeled the effect of an algorithm not being able to distinguish objects that are spaced closer than a certain cut-off distance. If the objects are located in a completely spatially random fashion, the resulting $L(r) - r$ function has an appearance similar to that observed in the analysis of the clathrin-coated pit data. These observations indicate that the resolving capabilities of image analysis algorithms need to be taken into consideration when analyzing object-based imaging data.

Importantly, this analysis also suggests that the resolving capabilities of an image processing approach can be characterized by the deviation from the expected spatial analysis results for objects that are simulated with a completely spatially random location pattern. We therefore determine the algorithmic resolution limit, α, of a particular object-based image analysis algorithm by using this algorithm to estimate the locations of objects that are simulated with completely spatially random positions. It is shown in Supplementary Note 5 (with further discussion in Supplementary Note 12) that resolution effects up to a distance of $\alpha$ impact the pair-correlation function of

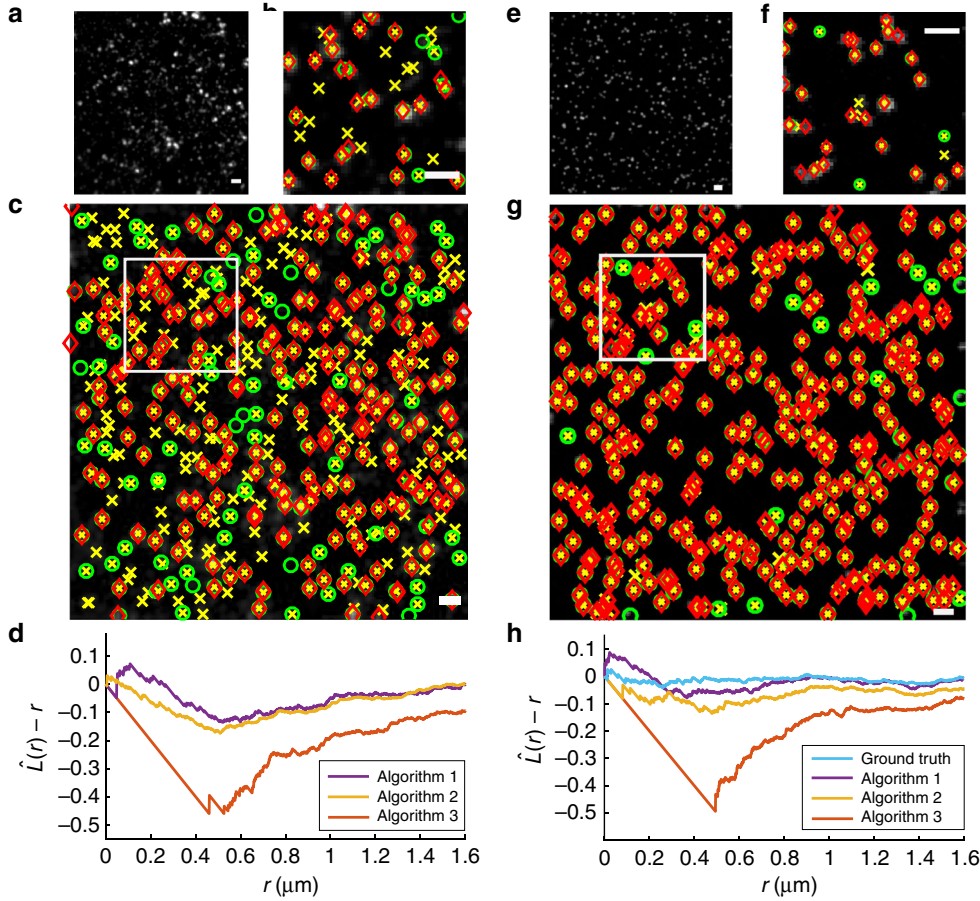

**Fig. 1** Detecting the effect of algorithmic resolution. **a** Fluorescence microscopy image of clathrin-coated pits on the membrane of an HMEC-1 cell. Scale bar = 1 μm. **b** Magnified view of the region marked in (**c**). Scale bar = 1 μm. **c** Location estimates obtained by applying three image analysis approaches to (**a**): Algorithm 1 (diamonds), Algorithm 2 (crosses), and Algorithm 3 (circles). Scale bar = 1 μm. **d** $\hat{L}(r) - r$ plots calculated based on the localizations shown in (**c**) appear to indicate that clathrin-coated pits are not distributed in a completely spatially random manner since the $\hat{L}(r) - r$ plots deviate significantly from 0 for each of the analysis approaches shown in (**c**). **e** Simulated image of clathrin-coated pits located at completely spatially random locations. Experimental and imaging parameters similar to (**a**) were used for the simulation: pixel size = 6.45 μm × 6.45 μm, magnification = 63, and background = 100 photons per pixel. Each clathrin-coated pit was simulated using a Gaussian profile with $\sigma = 120$ nm and total photon count uniformly distributed between 500 to 2000 photons. A total of 419 clathrin-coated pits were simulated in a 200 × 200 pixel image. Scale bar = 1 μm. **f** Magnified view of the marked region in (**g**). Scale bar = 1 μm. **g** Location estimates obtained using the image analysis approaches shown in (**c**) applied to (**e**). Scale bar = 1 μm. **h** $\hat{L}(r) - r$ plots calculated based on the localizations shown in (**g**) also results in significant deviations from 0 for a completely spatially random distribution of locations

completely spatially random data for distances up to 2α. Therefore, the algorithmic resolution limit α is then defined as half the distance in the pair-correlation function of the estimated object locations beyond which the graph exhibits a constant plot with value 1.

**Algorithmic resolution limit**. We analyzed several well-established image analysis algorithms. The purpose of this manuscript is not the evaluation of specific algorithms but to illustrate the methodology surrounding algorithmic resolution. We have largely used the algorithms with the default settings as they are available when downloaded. It is very well possible that an expert user of a particular algorithm could obtain significantly better results than those presented here.

We found that across the algorithms used, the algorithmic resolution limits can vary significantly (Fig. 2d). In fact, some of these algorithms are affected by algorithmic resolution well beyond the resolution limit that is predicted by Rayleigh's criterion, which is around 250–300 nm for the imaging conditions in Fig. 1. Algorithm 2 has the smallest algorithmic resolution limit

of 360 nm, whereas Algorithm 3 has an algorithmic resolution limit of 620 nm, almost twice that of Algorithm 2. Using completely spatially random data as a basis to analyze the resolution capability and to define the algorithmic resolution limit of object-based image analysis algorithms allows us to probe random configurations of object locations. Therefore, the concept of the algorithmic resolution limit also has applicability to object arrangements that are nonstochastic. As illustrated in Fig. 3a (see also Supplementary Figure 12), the deterministically arranged object locations that could be reliably identified coincide with those locations that are spaced at a distance larger than the algorithmic resolution limit.

Our analysis has also revealed shortcomings in some established algorithms beyond the impact of algorithmic resolution. Two of the algorithms, Algorithms 4 and 5, exhibit oscillatory behavior in the pair-correlation function even for very large distances. Upon further investigation, we found that these algorithms preferentially identify objects located towards the center of the pixels (see Supplementary Figure 11). As a result, the algorithmic resolution limit of these algorithms is taken as infinite or not defined. We also analyzed a multiemitter

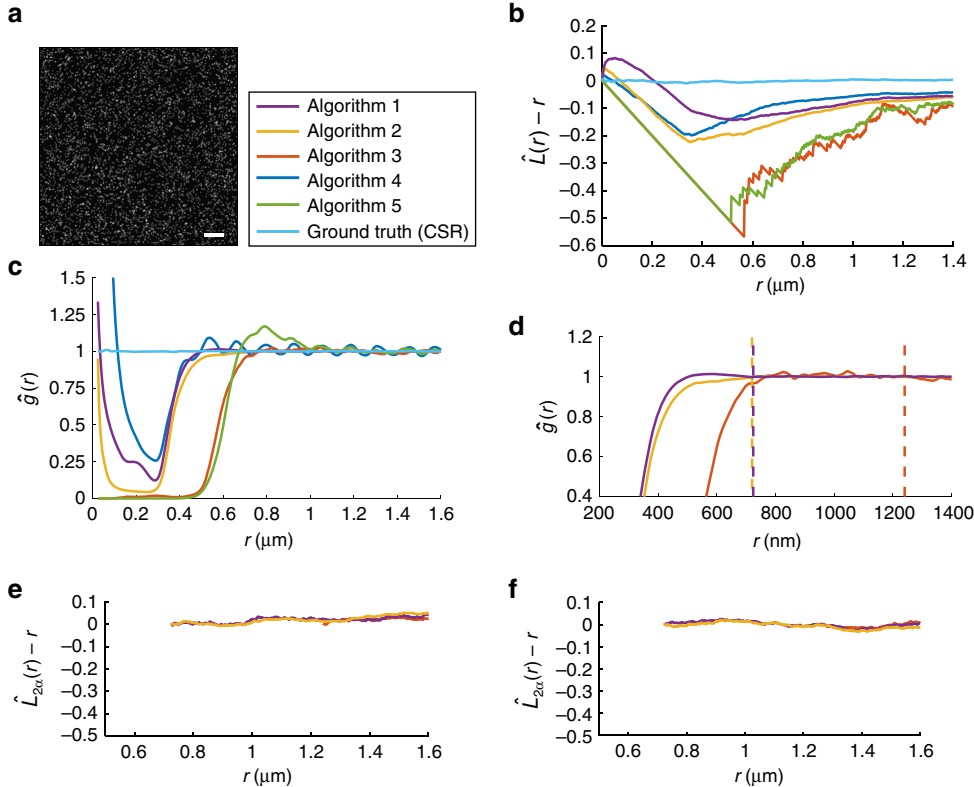

**Fig. 2** Determining the algorithmic resolution limit. **a** A sample simulated image of the dataset analyzed to obtain the results shown in (**b**) and (**c**). Each image consists of 2500 molecules positioned at completely spatially random locations over a 50 μm × 50 μm region. The following numerical parameters were used to generate each image: pixel size = 13 μm × 13 μm, magnification = 100, numerical aperture = 1.3, wavelength = 525 nm. Each molecule was simulated using an Airy profile with a total of 1000 photons. Scale bar = 5 μm. **b** $\hat{L}(r) - r$ plots calculated based on localizations obtained from various image analysis approaches applied to (**a**) exhibit different behaviors indicating different resolving capabilities. **c** Pair-correlations calculated based on the localizations obtained using the image analysis approaches shown in (**b**) applied to a dataset containing 2000 images generated similar to (**a**). These results are used to estimate the algorithm resolution limit $\hat{\alpha}$ (see Supplementary Note 9). The estimated algorithm resolution limits are as follows: $\hat{\alpha} = 362$ nm for Algorithm 1, $\hat{\alpha} = 360$ nm for Algorithm 2, and $\hat{\alpha} = 620$ nm for Algorithm 3. **d** Magnified view of the results shown in (**c**) with the values corresponding to $2\hat{\alpha}$ marked by dashed vertical lines. **e** Resolution-corrected $\hat{L}_{2\alpha}(r) - r$ plots calculated based on the results for $\hat{L}(r) - r$ shown in Fig. 1d and corrected using the $2\hat{\alpha}$ values shown in (**d**). **f** Resolution-corrected $\hat{L}_{2\alpha}(r) - r$ plots calculated based on the results for $\hat{L}(r) - r$ shown in Fig. 1h and corrected using the $2\hat{\alpha}$ values shown in (**d**) no longer show significant deviations from 0 for a completely spatially random distribution of locations

algorithm, Algorithm 6. Multiemitter algorithms have been introduced to deal with the location estimation of very closely spaced emitters. The corresponding pair-correlation function shows a very interesting profile that has a significantly more complex appearance than the pair-correlation functions for the other algorithms. For example, for low distances large values are obtained suggesting a clustering type behavior. Only for very large distance values does the pair-correlation function approach 1 (see Supplementary Figure 13a) which is also reflected in a very large algorithmic resolution limit of 577.5 nm which is significantly larger than the algorithmic resolution limit for the single emitter version. This emphasizes the difficulty in devising algorithms that can successfully resolve multiemitters and not introduce problems, although other multiemitter algorithms would need analyzing individually to check whether they exhibited similar behavior. The ring figures (see Supplementary Figure 13c-d) also reveal somewhat surprising behavior by overestimating the number of emitters for several isolated spots. This suggests that the algorithm, for the default settings we used, does not necessarily faithfully determine the multiplicity of the point sources.

We note at this point that the procedure used for estimating the algorithmic resolution limit from a pair-correlation function,

as described in Supplementary Note 9, is just one of many potential approaches. We have developed this automated method for consistency in our results, although some users may find that a visual inspection of the pair-correlation will suffice in understanding an algorithm's performance and behavior. As with any statistical procedure, the estimate is data dependent and will not be exact. To characterize this uncertainty, we have also proposed a bootstrapping procedure, as outlined in Supplementary Note 9. This uses a resampling-with-replacement strategy for creating several versions of the pair-correlation function and estimating the algorithmic resolution limit on each one. It is then possible to reason about the uncertainty of the estimator through analyzing the distribution of these bootstrapped estimates and computing approximate confidence intervals.

Ram et al.[12] analyzed the resolution of objects as it depends on the statistics of the acquired data, in particular the number of acquired photons from the objects and the various noise sources. It was shown that how well the distance between two objects can be measured, in terms of standard deviation, depends strongly on both the distance and the detected photons from the two objects. It is therefore expected that algorithmic resolution also depends on the specifics of the signal levels that are being used. This is indeed the case as illustrated in Supplementary Figure 9 where we

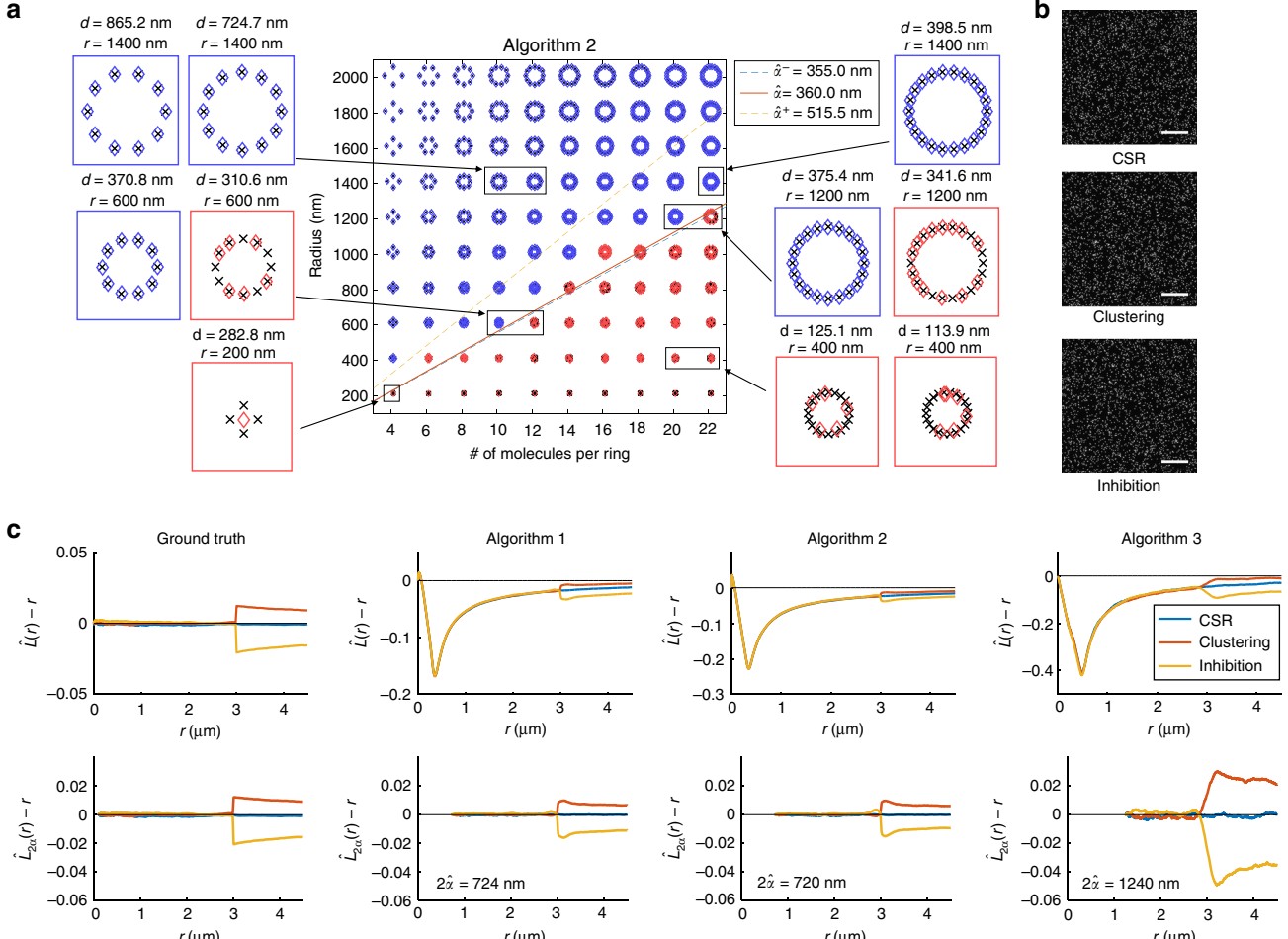

**Fig. 3** Application of algorithmic resolution limit to localization microscopy. **a** Application of the algorithmic resolution limit to the analysis of nonstochastic data illustrated using images of deterministic structures. Each structure consists of single molecules positioned evenly around the edge of a ring (crosses). Localizations were obtained by analyzing the image corresponding to each structure using Algorithm 2 (diamonds). Localizations corresponding to structures where all constituent molecules were accurately identified and localized to within 10 nm of the true location are shown in blue. Localizations corresponding to structures where one or more molecules were either not identified or where the localization deviated by more than 10 nm from the true location are shown in red. Magnified views of some structures are shown with the radius of the corresponding ring ($r$) and the distance between adjacent molecules on the edge of the ring ($d$) indicated above each magnified view. All molecules of structures where the spacing between adjacent molecules is greater than the algorithmic resolution limit of Algorithm 2 are accurately identified and localized to within 10 nm of the true location. The solid line corresponding to $\hat{\alpha} = 360\,\text{nm}$ indicates the algorithmic resolution limit for Algorithm 2. The dashed lines on either side of the solid line indicate the bootstrapped 80% confidence interval for the estimate of $\alpha$ (see Supplementary Note 9). Results obtained by analyzing the same images using other approaches are provided in Supplementary Figure 12. **b** Sample images from three datasets that were analyzed to obtain the results shown in (**c**). The three datasets were generated with the following spatial distribution of molecules (see Methods): completely spatially random (CSR) distribution, random distribution with a preferred spacing ranging from 2990 to 3010 nm between molecules (clustering), and random distribution with molecules avoiding spacings between 2990 to 3010 nm of each other (inhibition). Scale bar = 10 μm. **c** $\hat{L}(r) - r$ plot compared to the corresponding resolution-corrected $\hat{L}_{2\alpha}(r) - r$ plot calculated based on localizations obtained by analyzing the three datasets illustrated in (**b**). For each analysis approach, the value corresponding to $2\hat{\alpha}$ indicated in Fig. 2d is used to calculate $\hat{L}_{2\alpha}(r)$. Results show that deviations from 0 in the $\hat{L}(r) - r$ plot are corrected for completely spatially random distributions of locations when the algorithmic resolution limit is taken into account. Results for distributions with clustering or inhibition spacings between molecules still show corresponding deviations from 0 in the corrected $\hat{L}_{2\alpha}(r) - r$ results

see that for simulations with extremely low signal levels the pair-correlation functions for the different algorithms take on different forms and as a result also worsen the algorithmic resolution limit. Importantly, however, for all other higher signal levels such as those for which single-molecule experiments are usually conducted, there are no appreciable differences in the regions of the curves that determine the algorithmic resolution limit and as a result in the estimate of the limit itself. This independence is further consistent with the results of Ram et al.[12,13] as for the distances around Rayleigh's resolution criterion very good estimates can be obtained even for relatively low signal levels.

**Quantifying the effect of algorithmic resolution**. The question immediately arises, how the algorithmic resolution limit of an image analysis algorithm impacts the analysis of experimental data. For example, it is important to quantitate how many objects remain unaffected by resolution effects when the imaging data are analyzed using an algorithm with algorithmic resolution limit $\alpha$. As predicted by the theory and verified empirically, the algorithmic resolution limit of an algorithm is independent of the density of objects within the density ranges of general interest (see Supplementary Note 10 and Supplementary Figure 8). However, understanding how the algorithmic resolution limit, object

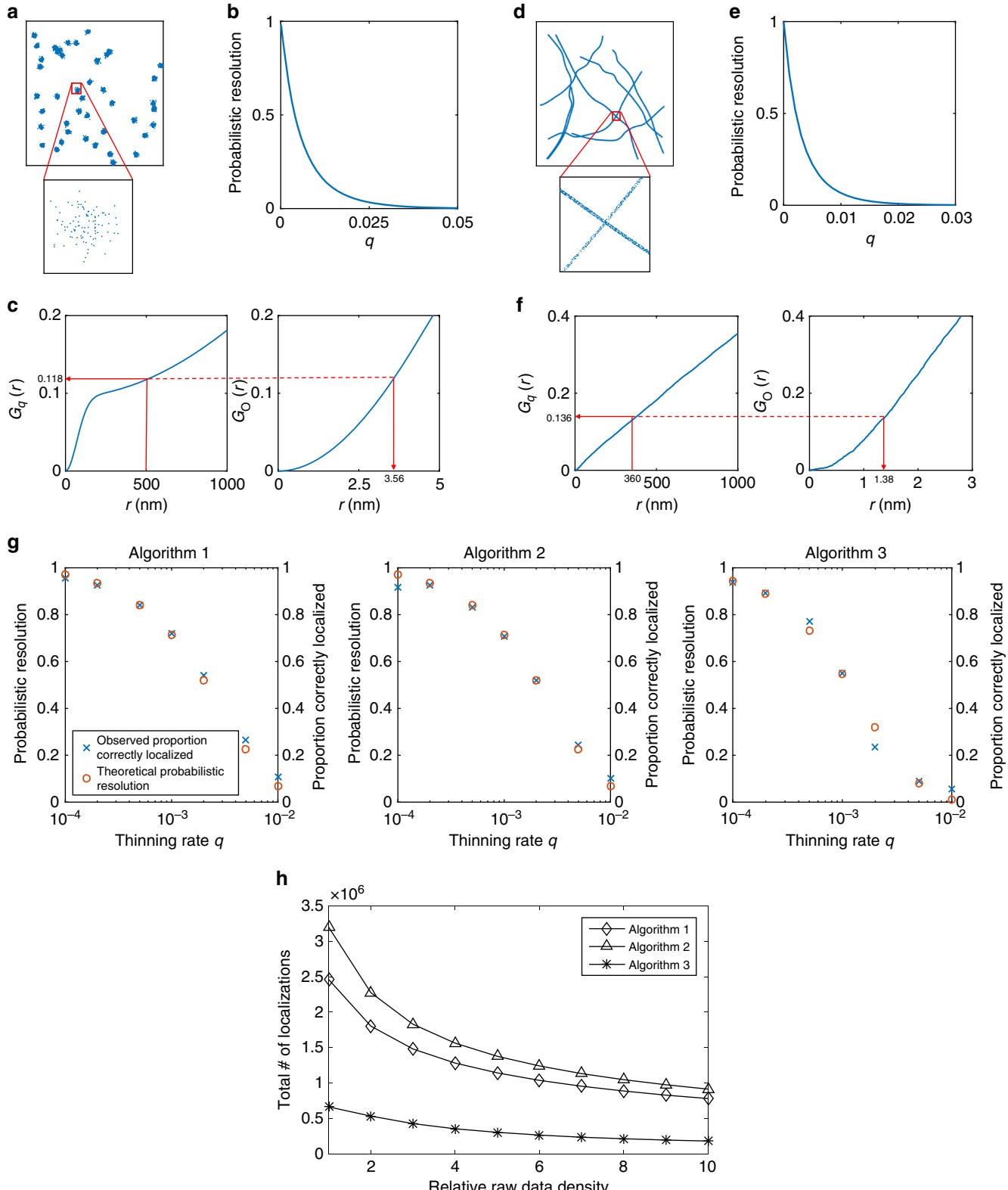

density, and spatial structure of the objects interplay with one another is of vital importance. The effect of object density in fluorescence microscopy is of course well studied[14–17] and here we provide further insight into its role.

As shown in Supplementary Note 6, the probability that an object is unaffected by resolution effects, what we here call the probabilistic resolution, is given by $1 - G_O(\alpha)$. Here, $G_O$ is the nearest-neighbor probability distribution function for the objects, describing the probability that an arbitrary object is at a distance less than $\alpha$ from its nearest neighboring object (see Supplementary Note 2 for its formal definition). If the objects are located according to a completely spatially random distribution, $G_O(\alpha) = 1 - \exp(-\lambda_O \pi \alpha^2)$, where $\lambda_O$ is the density of the object locations. Therefore, in a highly dense object pattern, not surprisingly, the

**Fig. 4** Probabilistic resolution. **a** Realization of a clustered spatial point pattern, with an expected 10 clusters per 30 µm², an expected 100 objects per cluster, each distributed around the cluster center with a standard deviation of 0.05 µm in both $x$ and $y$ directions. **b** The probabilistic resolution as a function of $q$, the probability of an object appearing in any given frame, for the clustered spatial point pattern shown in (**a**). **c** $G_O(r)$, the nearest-neighbor distribution function for the clustered spatial point pattern shown in (**a**), and $G_q(r)$, the nearest-neighbor distribution function for a random subset of points (replicating a single frame in a localization microscopy experiment) where the probability of an object appearing is $q = 1/1000$. The value of $G_q$ at an example algorithmic resolution limit of $\alpha = 500$ nm is shown to be 0.118, which gives a probabilistic resolution (the probability an object is unaffected by resolution) of 88.2%. The algorithmic resolution that would give the same probabilistic resolution when all objects are imaged in a single frame, given as $G_O^{-1}(G_q(\alpha))$, is shown to be 3.56 nm. **d** Tubulin spatial point pattern[10]. **e** The estimated probabilistic resolution of the tubulin data shown in (**d**) as a function of $q$, the probability of an object appearing in any given frame. **f** $G_O(r)$, the estimated nearest-neighbor distribution function for the clustered spatial point pattern shown in (**d**), and $G_q(r)$, the estimated nearest-neighbor distribution function for a random subset of points (replicating a single frame in a localization microscopy experiment) where the probability of an object appearing is $q = 1/2401$. The value of $G_q$ at an example algorithmic resolution limit of $\alpha = 360$ nm is shown to be 0.136, which gives a probabilistic resolution (the probability an object is unaffected by resolution) of 86.4%. The algorithmic resolution that would give the same probabilistic resolution when all objects are imaged in a single frame, given as $G_O^{-1}(G_q(\alpha))$, is shown to be 1.38 nm. **g** The theoretical probabilistic resolution of the tubulin data for Algorithms 1, 2, and 3 at different values of $q$ is shown with red circles. The blue crosses show the observed proportion of correctly localized molecules. This was determined to be ground truth coordinates that have a localization within 12 nm. **h** The number of localizations estimated by Algorithms 1, 2, and 3 on experimental tubulin data as a function of (relative) data density (see text for details)

probability that an object is not affected by algorithmic resolution is severely reduced.

For example, consider cellular membrane receptor clusters distributed in a completely spatially random fashion with a density of 1 cluster per square micrometer (as in Fig. 1e). When analyzing the location of such protein clusters using an algorithm with algorithmic resolution limit $\alpha = 360$ nm (e.g., Algorithm 2), the probability a cluster is not affected by resolution (the probabilistic resolution) is 66.5%. However, when analyzing the cluster locations using an algorithm with $\alpha = 620$ nm (e.g., Algorithm 3), the probabilistic resolution is 29.9%. Thus, the difference in algorithmic resolution between the two algorithms can have drastic effects on the analysis of the data. Further, it is only for cluster densities of 0.1 clusters per square micrometer that the probabilistic resolution will be above 95% (with 96.0%) for an algorithm with $\alpha = 360$ nm. However, this probability decreases significantly to 88.6% for the algorithm with $\alpha = 620$ nm. Other spatial distributions can be assessed through similar analysis of their nearest-neighbor distribution function. This will be demonstrated by applying this probabilistic approach to localization microscopy.

**Application to localization microscopy.** Localization-based superresolution methods use repeat stochastic excitation of small subsets of the fluorophores in a sample[4]. The question therefore arises how small these subsets need to be in order for a large fraction of the single molecules/objects to be spatially isolated and not affected by the algorithmic resolution limit of the analysis step.

To quantify this, let us denote $q$ to be the probability of an object appearing in any given frame of the dataset. For example, in an ideal PALM experiment where molecules are evenly distributed across $n_F$ frames, we have $q = n_F^{-1}$, and in the classical microscopy setting, when all objects appear in a single frame, we have $q = 1$. It is shown in Supplementary Note 7 that the probabilistic resolution becomes $1 - G_q(\alpha)$. Here, $G_q$ is the nearest-neighbor distribution function for the subset of objects that appear in an arbitrary frame. We recall that $G_q(\alpha)$ is the probability an arbitrary object within the frame is within a distance $\alpha$ of its nearest neighbor. Therefore, for $0 < q < 1$, we have $G_q(\alpha) < G_O(\alpha)$, where $G_O$ is the nearest-neighbor distribution function for the full dataset of objects (i.e. if all objects appear in a single frame). This shows how the probabilistic resolution increases by separating the objects among frames. Furthermore, if we increase the number of frames from $n_F$ to $n_F'$, and hence decrease the probability an object appears in an arbitrary frame from $q$ to $q'$, we have $G_{q'}(\alpha) < G_q(\alpha)$ and the probabilistic

resolution further increases. Importantly, it is shown in Supplementary Note 7 that probabilistic resolution becomes 1 (i.e. all objects are perfectly resolved) as $q = n_F^{-1}$ tends to zero. More so, if one wishes to insist the probabilistic resolution must be greater than some value $p$, it is required that $q$ is chosen such that $p < 1 - G_q(\alpha)$. We will illustrate these concepts with two examples: clustered objects and tubulin data.

**Clustered objects.** Suppose the single molecules to be localized exist in clusters. A simple model one may apply in this setting is that the cluster centers are completely spatially random and single molecules are distributed about the cluster centers according to a 2D spherically symmetric Gaussian distribution. In this setting, the nearest-neighbor distribution function can be derived[18] and is presented in Supplementary Note 8. For example, consider imaging a clustered process of this type in which there is an average of ten clusters per 30 µm², an expected 100 single molecules per cluster, each distributed around the cluster center with a standard deviation of 0.05 µm in both $x$ and $y$ directions. An example realization of this process is shown in Fig. 4a. For an algorithmic resolution limit of $\alpha = 500$ nm, if all the single molecules were to be imaged in a single frame the probability of an arbitrary object being unaffected by resolution is 0 (to machine precision). However, if the single molecules were imaged using sparse subsets across multiple frames with a probability of $q = 1/1000$, the probabilistic resolution rises to 88.2%. For $q = 1/10,000$ it rises further to 96.8%. Suppose we wish to achieve a probabilistic resolution of at least 99% we require $q \leq 7.97 \times 10^{-5}$, which would require a minimum of 12,548 frames to achieve. Completely spatially random molecules of the same average intensity would require only 3907 frames ($q = 2.56 \times 10^{-4}$) to attain the same level of probabilistic resolution. This demonstrates the extra demands that clustered data present.

Interestingly, it is shown in Supplementary Note 7 that in order to achieve the same level of probabilistic resolution in a classical single-molecule experiment where all single molecules are activated and imaged in a single acquisition, an algorithmic resolution of $G_O^{-1}(G_q(\alpha))$ is required. When $q = 1/1000$ and $\alpha = 500$ nm, this would require an algorithmic resolution limit of 3.56 nm, which is well beyond what is currently achievable, thus illustrating the power of using stochastic excitation for localization-based single-molecule superresolution experiments. A demonstration of this calculation is shown in Fig. 4c.

**Tubulin data.** We consider a tubulin dataset[10] consisting of $1 \times 10^5$ emitters imaged over 2401 frames (see Fig. 4d). This gives

$q = 1/2401$ and an average of 41.6 emitters per frame. Supplementary Note 8 gives details on computing the nearest-neighbor distribution function for this dataset. For this value of $q$ and an algorithmic resolution limit of $\alpha = 360$ nm (e.g., Algorithm 2), the probabilistic resolution equal to $1 - G_q(\alpha)$ is found to be 86.4%. To achieve the same level of probabilistic resolution in a classical single-molecule experiment where all single molecules are activated and imaged in a single acquisition, an algorithmic resolution of 1.38 nm is required, again illustrating the power of using stochastic excitation for localization-based single-molecule superresolution experiments. A demonstration of this calculation is shown in Fig. 4f. Changing $q$ to 1/10,000 increases the probabilistic resolution to 96.8% whereas for $q = 1/1000$ the probabilistic resolution is decreased to 71.2%. Verification of this analysis is provided in Fig. 4g. The proportion of correct localizations (within 12 nm of a ground truth coordinate) is plotted for different values of $q$. This is compared to the theoretical probabilistic resolution as presented in Supplementary Note 7 and estimated using the scheme in Supplementary Note 8.

An experimental tubulin dataset is also considered to further verify the results presented. The original dataset, Dataset 1, is comprised of 50,000 frames and contains approximately $3.2 \times 10^6$ localizations for Algorithm 1, $2.5 \times 10^6$ localizations for Algorithm 2 and $0.7 \times 10^6$ localizations for Algorithm 3. Averaging pairs of images, we are able to create Dataset 2 that consists of 25,000 frames, each with double the object density. Dataset 3, formed by averaging triplets of frames, consists of 16,666 frames with triple the object density. This is repeated up to Dataset 10, which consists of 5000 frames with ten times the object density. The number of localizations generated by each algorithm for each dataset is shown in Fig. 4h (and Supplementary Figure 14 shows examples of the reconstructed images from these datasets). These results demonstrate two key points. The first is it can be seen that Algorithm 2, the algorithm with the smallest resolution limit, has the largest number of localizations. Furthermore, Algorithm 1, which has a similar algorithmic resolution limit, has a very similar number of localizations. However, Algorithm 3, which has an algorithmic resolution limit almost twice as large as Algorithms 1 and 2, produces far fewer localizations. This is consistent with the presented theory; a larger algorithmic resolution results in a smaller probabilistic resolution under the same molecule density, and as such we would expect fewer localizations. The second point is that the number of localizations decreases as the density increases. Again, this is predicted under our theoretical framework since we show probabilistic resolution decreases with density. This demonstrates that experimental observations on the performance of different algorithms are consistent with the findings of the paper, which have been reached from simulation methods.

**Adjusting for algorithmic resolution**. We have seen that algorithmic resolution can significantly distort Ripley's $K$-function. However, knowing the algorithmic resolution limit $\alpha$ of an algorithm allows us to define a resolution-corrected Ripley's $K_{2\alpha}$-function and resolution-corrected $L_{2\alpha}(r) - r$ for $r \geq 2\alpha$ (see Supplementary Note 11). Figure 3c shows that inhibition and clustering can be correctly identified with the resolution-corrected $L_{2\alpha}(r) - r$ function if they occur at distances above $2\alpha$ for object-based imaging data analyzed with an algorithm of resolution limit $\alpha$.

If the clathrin-coated pit data of Fig. 1a is analyzed using an algorithm with algorithmic resolution limit $\alpha = 360$ nm (e.g., Algorithm 2) and the estimated locations processed with the resolution-corrected $L_{2\alpha}(r) - r$ function, the data show that there is no significant deviation from complete spatial randomness

beyond the distance of $2\alpha = 720$ nm (Fig. 2e, f). This indicates that at distances above twice the algorithmic resolution limit for the individual algorithms, the clathrin-coated pit locations do not show any deviation from complete spatial randomness.

## Discussion

Resolution has been analyzed in microscopy going back to the classical criteria by Rayleigh and Abbe. Those criteria address the performance of the imaging optics. Using an information-theoretic approach, Rayleigh's resolution criterion was generalized and put in the context of modern imaging where data consist of noise-corrupted photon count measurements acquired through quantum-limited detectors[12]. A resolution measure based on the Fourier ring coefficient was introduced that can be computed directly from an acquired image and takes into account the standard deviation with which a single molecule can be localized[19]. Common to these recent approaches is they do not take into account that different object-based image analysis algorithms can have very different algorithmic resolution limits.

The evaluation of algorithms for single-molecule image analysis is complex and a number of approaches have been used in the past, many of those comparing the estimated locations with the ground truth of simulated data[10]. The introduction of the concept of algorithmic resolution in this paper provides an additional tool by determining the minimum distance beyond which the algorithm can reliably distinguish different objects. The analysis presented here also provides important insights into why algorithms perform differently on more classical evaluation approaches (see e.g. Fig. 3a).

Object density has been well recognized as causing significant problems for the object-based image analysis algorithms[14–17]. Here, we have derived analytical approaches that investigate how algorithmic resolution and object density impact the probability of an object being affected by resolution when the data are analyzed with an image analysis algorithm with specific algorithmic resolution limit. Understanding this probabilistic resolution is key to determining if post-processing methods, for example clustering algorithms[20] that estimate cluster sizes and the number of objects per cluster, can be used with confidence.

The approach to defining algorithmic resolution that is presented here depends on the availability of test data with objects that are distributed in a completely spatially random fashion. Such data are completely uncorrelated and consequently gives rise to a pair-correlation function that is identically equal to 1. The estimation algorithm that is investigated is then used to estimate locations of these objects based on simulated data with the given object locations. Subsequently, an estimated pair-correlation function is computed based on the estimated locations. The algorithmic resolution limit is defined through the distance below which the estimated pair-correlation function deviates from that based on the true locations, i.e. 1. This approach could be modified to use test data that gives rise to a different pair-correlation function. The point at which the estimated pair-correlation function deviates from the theoretical one could be used as a method of comparing two or more different algorithms. However, with the theoretical results of Supplementary Note 5 pertaining to completely spatially random data, caution should be taken in using these points to estimate algorithm resolution limits.

For the approach to be relevant to experimental settings, it is important that the simulated data used to determine the algorithmic resolution limit reflects the data that will be acquired when the algorithm is applied in an experimental setting. As we have seen, the determination of the algorithmic resolution limit does depend on the photon count of the simulated single-molecule data in certain circumstances, such as extremely low

photon counts. We have also seen a dependence on the density of the sampled objects, again in extreme cases. It is equally clear that other model parameters that determine the appearance of the images of the objects under study need to be matched to the experimental settings to be able to expect reliable results. If experimental data were available that can be guaranteed to be made up of objects that are located in a completely spatial random manner, then such data could also be used. However, guaranteeing that objects are located in a completely spatially random fashion would be very difficult to achieve. Unless such a guarantee is available, simulated data are to be preferred as complete spatial randomness of these test data is critical for the approach.

We have introduced a methodology to systematically assess the algorithmic resolution limit of object-based image analysis algorithms and to evaluate the impact of the limitations on the analysis of microscopy data. We hope that the approaches presented will contribute to a systematic evaluation of such algorithms that are of relevance not only to microscopy applications but to other object-based imaging scenarios such as those arising, for example, in astronomy.

Spatial statistics has played an important role in many areas of cell biology. We hope that the results presented here will contribute to an improved understanding of the methodology and lead to the avoidance of misinterpretation of the acquired data. In particular the use of the resolution-corrected Ripley's K-function is expected to be a powerful tool for cell biological studies that rely on spatial statistics.

## Methods

**Preparing HMEC-1 cells for fluorescence imaging.** HMEC-1 cells were fixed using 1.7% (w/v) paraformaldehyde (Electron Microscopy Sciences) at room temperature and permeabilized by incubation with 0.02% (w/v) saponin in phosphate-buffered saline (PBS) for 10 min at room temperature. Cells were then preblocked with 3% bovine serum albumin (BSA) in PBS, incubated with anti-Clathrin primary antibody (mouse monoclonal X22, Catalog # ab2731, Abcam, diluted 1000-fold in 1% BSA/PBS) for 25 min at room temperature, and treated with goat serum diluted 50-fold. Bound primary antibody was detected by treatment with Alexa 555-labeled anti-mouse IgG (Catalog # A-21424, Thermo Fisher Scientific, diluted 750-fold in 1% BSA/PBS) for 25 min at room temperature. Cells were washed twice with PBS between each incubation and finally immersed in 1.5 ml of 1% BSA/PBS prior to imaging.

**Fluorescence microscopy imaging.** Fixed HMEC-1 cells were imaged with a Zeiss (Axiovert 200M) inverted epifluorescence microscope fitted with a 63× (1.4 NA) Plan Apo objective (Carl Zeiss) using a CCD camera (Orca ER, Hamamatsu). The sample was illuminated using a broadband LED illumination (X-Cite 110LED, Excelitas Technologies) filtered through a standard Cy3 filterset (Cy3-4040C-ZHE M327122 Brightline, Semrock). Signal from the sample was also filtered through this filterset before being acquired by the camera.

**Preparing BS-C-1 cells for superresolution imaging.** BS-C-1 cells (CCL-26, American Type Culture Collection) were plated on custom, glass-bottom dishes (Catalog # PG35G-10C-NON, MatTek Corporation) fitted with high-performance Zeiss coverglasses (Catalog # 474030-9000-00, Thickness 1.5) at a density of 20,000 cells and incubated for 16–24 h. The coverglasses were cleaned by sonicating them sequentially in 50% HPLC grade ethanol, 1 mM HCL in 50% HPLC grade ethanol, and 1 M KOH in 50% HPLC grade ethanol for 20 min each with extensive washing using MilliQ water after every sonication. The cells were fixed using 3.4% paraformaldehyde (Electron Microscopy Sciences) in PBS for 10 min at 37 °C. The fixed cells were then permeabilized using 0.1% Triton X-100 in PBS for 10 min at room temperature. Cells were then preblocked with 5% BSA (Catalog # BP1600, Fisher Scientific), incubated with mouse α-tubulin antibody (Catalog # A11126, Thermo Fisher Scientific, diluted 200-fold in 1% BSA/PBS) for 30 min at room temperature, and treated with goat serum (Catalog # G6767, Sigma, diluted 50-fold in 1% BSA/PBS). Bound primary antibody was counter-stained with Alexa 647-labeled goat anti-mouse IgG (Catalog # A-21236, Thermo Fisher Scientific, diluted 750-fold in 1% BSA/PBS) for 30 min at room temperature.

**Optical setup for superresolution imaging.** Images of BS-C-1 cells were acquired using a standard inverted epifluorescence microscope (Observer A1, Zeiss) fitted with a 63× (1.4 NA) Plan-Apochromat oil-immersion objective (Carl Zeiss). The

sample was illuminated using 635 nm and 450 nm diode lasers (OptoEngine) for the excitation of Alexa 647 and for photoswitching respectively. The illumination light was directed towards the microscope using custom laser excitation optics, reflected into the sample using a dichroic filter (Di01-R405/488/561/635-25x36, Semrock), and focused onto the back focal plane of the objective lens. Signal from the sample was filtered using a bandpass filter (FF01-676/29-25, Semrock) and acquired by an electron multiplying charge coupled device (EMCCD) camera (iXon DU897-BV, Andor) set to conventional readout mode. Axial drift during the course of the acquisition was corrected in real time using a custom focus stabilization system comprising an 850 nm diode laser (PI, USA), a quadrant position detector (Thorlabs), and an XYZ piezo positioner (PI, USA). All devices including lasers, shutters, and cameras were controlled and synchronized using custom-written software in the C programming language.

**Superresolution image acquisition.** Buffer solutions in which the BS-C-1 cell samples were prepared were replaced with an imaging buffer immediately prior to the imaging of each sample. This imaging buffer consisted of 50 mM cysteamine, 0.5 mg ml$^{-1}$ glucose oxidase, 40 μg ml$^{-1}$ catalase, and 10% glucose (w/v) in PBS, pH 7.4. (All buffer ingredients were purchased from Thermo Fisher Scientific.) The imaging buffer was freshly prepared for each acquisition. A coverslip was placed over the sample and sealed using vacuum grease. For each acquisition, the sample was briefly illuminated (approximately 5–10 min) using the maximum-power setting of the 635 nm laser to switch off most of the Alexa 647 fluorophores and achieve sparse single-molecule activation conditions. Acquisition was initiated once steady-state, single-molecule blinking was observed with periodic illumination using the 405 nm laser to reactivate fluorophores and maintain sufficient activation density. The activation laser power was kept low throughout the acquisition. For each dataset, typically 50,000 images of stochastically activated fluorophores were acquired with 50 ms exposure time for each image and without EM gain resulting in an acquisition rate of 20 frames per second.

**Generating completely spatially random locations.** For a completely spatially random distribution of locations in an $S\,\mu m \times S\,\mu m$ region, the $(x, y)$ coordinate for each location was obtained by drawing realizations of independent random variables $X$ and $Y$, each uniformly distributed with probability density functions $p_X(x) = p_Y(y) = 1/S$, $0 \le x, y \le S$. For the simulated image in Fig. 2, $S = 50$ resulting in a 50 μm × 50 μm region.

**Generating locations for deterministic structures.** Deterministic structures consist of $D$ molecules located at evenly spaced points on the circumference of a circle of radius $r$. The location of the $d$th point $(x_0^d, y_0^d)$ is given by $x_0^d = r \cos(2\pi d/D)$ and $y_0^d = r \sin(2\pi d/D)$, where $d = 1,\ldots,D$.

**Generating locations with a preferred spacing (clustering).** A set of $N$ random locations, denoted by $\Delta$, such that $N_c$ pairs of those locations are spaced at distances between $r_{min}$ and $r_{max}$ is generated by combining three subsets of locations so that $\Delta = \Delta^c \cup \Delta^{c'} \cup \Delta^u$. The subset $\Delta^c := \{d_1^c, \ldots, d_{N_c}^c\}$ consists of completely spatially random locations $d_i^c := (x_i^c, y_i^c)$. The subset $\Delta^{c'} := \{d_1^{c'}, \ldots, d_{N_c}^{c'}\}$ consists of locations corresponding to $\Delta^c$, where each location $d_i^{c'} := (x_i^{c'}, y_i^{c'})$, is calculated as

$$x_i^{c'} = x_i^c + r_i \cos\theta_i,$$
$$y_i^{c'} = y_i^c + r_i \sin\theta_i.$$

The distance $r_i$ between $d_i^c$ and $d_i^{c'}$ is uniformly distributed between $r_{min}$ and $r_{max}$, and $\theta_i$ is uniformly distributed between 0 and $2\pi$. The subset $\Delta^u$ is an additional completely spatially random distribution of $N_u = N - 2N_c$ locations. For the simulated images analyzed to obtain the results in Fig. 3, the following values were used: $N = 2500$, $N_c = 250$, $r_{min} = 2990$ nm, and $r_{max} = 3010$ nm.

**Generating locations avoiding specific spacings (inhibition).** A set of locations, $\Delta := \{d_1,\ldots,d_N\}$, in which no two locations are spaced between $r_{min}$ and $r_{max}$ of each other is generated as follows. For $i = 1,\ldots,N$, the $i$th location $d_i$ is drawn from a completely spatially random distribution. The $i$th point is not added as a location if $r_{min} \le d_{ij} \le r_{max}$ for some $1 \le j \le i$, where $d_{ij}$ denotes the distance between the $i$th and $j$th locations. For the simulated images analyzed to obtain the results in Fig. 3, the following values were used: $N = 2500$, $r_{min} = 2990$ nm and $r_{max} = 3010$ nm.

**Simulating an image of clathrin-coated pits.** When simulating an image of clathrin-coated pits, the detector is modeled as a set of pixels $\{C_1,\ldots,C_K\}$. The photon count detected in the $k$th pixel is modeled as $T_k := S_k + B_k$, where $S_k$ and $B_k$ are both Poisson random variables[21]. The total photons detected at the $k$th pixel from all clathrin-coated pits within the region represented by the image is denoted by $S_k$. The background photon count $B_k$ has a mean of $B = 100$ photons per pixel. The mean of $S_k$ is given by

$$\mu_k := \sum_{d=1}^{D} \left( N_d \int_{C_k} f_d(r) dr \right),$$

where $D$ is the total number of clathrin-coated pits in the image, $N_d$ denotes the total number of photons detected from the $d$th clathrin-coated pit, $C_k$ denotes the area of the $k$th pixel, and $f_d$ denotes the photon distribution profile for the $d$th clathrin-coated pit. For the simulated image of clathrin-coated pits in Fig. 1, $D = 419$ and $N_d$ had values uniformly distributed between 500 to 2000 photons.

For each clathrin-coated pit, $f_d$ is modeled as a Gaussian profile given by

$$f_d(r) := \frac{1}{2\pi M^2 \sigma^2} \cdot \exp\left( -\frac{(x - Mx_0^d)^2}{2(M\sigma)^2} - \frac{(y - My_0^d)^2}{2(M\sigma)^2} \right),$$

where $M$ denotes magnification, $\sigma$ denotes the width of the Gaussian profile, and $(x_0^d, y_0^d)$ denotes the center of the $d$th clathrin-coated pit. The coordinate $(x_0^d, y_0^d)$ is drawn from a completely spatially randomly distributed set of $D$ locations generated as described above. For the simulated image in Fig. 1, $M = 63$ and $\sigma = 120$ nm.

**Simulating images of single molecules.** When simulating an image of single molecules, the detector is again modeled as a set of pixels $\{C_1, \ldots, C_k\}$. The photon count detected at the $k$th pixel is modeled as a Poisson random variable with mean given by,

$$\mu_k := \sum_{d=1}^{D} N \int_{C_k} f_d(r) \mathrm{d}r + B,$$

where $N$ denotes the total number of photons detected from the molecule, $C_k$ denotes the area of the $k$th pixel, $f_d$ denotes the photon distribution profile for the $d$th molecule, and $B$ denotes the uniform number of background photons detected in each pixel. For each molecule, $f_d$ is modeled as an Airy profile given by

$$f_d(r) := \frac{\left[ J_1\left( \frac{\kappa}{M} ||r - r_0^d|| \right) \right]^2}{\pi ||r - r_0^d||^2},$$

where $J_1$ denotes the first-order Bessel function of the first kind, $||r - r_0^d|| = \left( (x - Mx_0^d)^2 + (y - My_0^d)^2 \right)^{1/2}$, $(x_0^d, y_0^d)$ denotes the location of the $d$th molecule, and $M$ denotes the magnification of the optical system. $\kappa$ is calculated as $\kappa = 2\pi N_a/\lambda$, where $N_a$ denotes the numerical aperture, and $\lambda$ denotes the wavelength of the photons emitted by the molecule. The following values were used for simulating all images of single molecules: $N_a = 1.3$, $\lambda = 525$ nm, $M = 100$, and $N = 1000$ photons. Images that were analyzed for all figures were simulated using a background of $B = 0$ photons per pixel.

**Simulating PALM images of tubulin.** When simulating images of fluorescently labeled tubulin molecules that are stochastically photoactivated and detected, each image is taken as an image of single molecules and is simulated as described above. The positions of molecules within each image are obtained by taking a spatial point pattern consisting of $N$ points and uniformly distributing them among $n_F = \lceil q^{-1} \rceil$ images, where $q$ is the probability that a molecule is visible in any particular frame. For all simulated tubulin datasets, the spatial point pattern provided by Sage et al.[10] was used.

**List of image analysis approaches.** The following is a list of the image analysis approaches that were used:

- **Algorithm 1 (Wavelet):** Detects molecules or clathrin-coated pits using wavelet-filtering[22] and estimates their locations by fitting Airy profiles to the detected molecules or Gaussian profiles to the detected pits. Further details are provided below.
- **Algorithm 2** Detects and localizes molecules using the default settings of the software package taken from ref. [23].
- **Algorithm 3** Detects and localizes molecules using the default settings of the software package taken from ref. [24].
- **Algorithm 4 (Global-Thresholding):** Detects molecules or clathrin-coated pits by identifying pixels above a threshold value and estimates their locations by fitting Airy profiles to the detected molecules or Gaussian profiles to the detected pits. Further details are provided below.
- **Algorithm 5** Detects and localizes molecules using the default settings of the software package taken from ref. [25].
- **Algorithm 6** Detects and localizes molecules using the default settings of the multiemitter version of the software package taken from ref. [23].

For Algorithms 2, 3, 5, and 6, the software packages provided by the authors of the algorithms were used. We used the default settings of those software packages and did not attempt to optimize them for use here. The purpose of this paper is not to investigate the different algorithms. We used several algorithms solely to illustrate that different algorithms can have different algorithmic resolutions limits. It is quite likely that an expert could obtain very different results from those shown here. We would therefore like to emphasize that we are not interested in an evaluation of these algorithms per se. To arrive at such an evaluation in a rigorous manner would require a very different approach. For example, the involvement of the authors of the software packages, as is routinely done in software evaluation

challenges, is the preferred approach. It is under these provisos that we somewhat reluctantly mention the references to the software packages that were used.

**Identifying objects by wavelet-filtering.** The image was filtered using the product of two consecutive wavelet transforms as described in ref. [22]. Each isolated set of one or more edge-connected pixels obtained from the filtering was identified as a region of the image containing an object, i.e., an individual molecule or clathrin-coated pit. For the subsequent localization of that object, a $5 \times 5$ pixel region centered on the average pixel coordinate of the corresponding set of identified pixels was used.

**Identifying objects by global-thresholding.** The image was thresholded using 25% of the maximum pixel intensity in the dataset as the threshold value. Each isolated set of one or more edge-connected pixels obtained from the thresholding was identified as a region of the image containing an object, i.e., an individual molecule or clathrin-coated pit. For the subsequent localization of that object, a $5 \times 5$ pixel region centered on the average pixel coordinate of the corresponding set of identified pixels was used.

**Localizing objects detected by wavelet/threshold approaches.** Each clathrin-coated pit or single molecule was localized by fitting a Gaussian or Airy profile, respectively, to the corresponding $5 \times 5$ pixel image identified using either wavelet-filtering or global-thresholding as described above. An initial location estimate and an initial value for the $\sigma$ parameter denoting the width of a Gaussian profile was calculated for each clathrin-coated pit or single molecule by applying the approach described in ref. [26] to the corresponding image. An initial value for the $\kappa$ parameter of the Airy profile was calculated as $1.323/\sigma$. The background associated with each clathrin-coated pit or single molecule was taken as the median of the intensities in the edge pixels of the corresponding $5 \times 5$ pixel image. An initial estimate of the photon count detected from each molecule or clathrin-coated pit was taken as the sum of the pixel intensities in the corresponding image after subtracting the background. Airy or Gaussian profiles with initial values for the various parameters calculated as described above were fitted to each $5 \times 5$ pixel image using a least-squares estimator to obtain the final location estimates. The location parameters $(x_0, y_0)$, width parameter ($\sigma$ when fitting Gaussian profiles and $\kappa$ when fitting Airy profiles), and the total photon count were estimated for each clathrin-coated pit or single molecule.

**Estimating Ripley's K-function from one image.** When estimating $L(r) - r$ for a set of localizations obtained by analyzing an image of either clathrin-coated pits or single molecules, $L(r) = \sqrt{K(r)/\pi}$ for $r > 0$. $K(r)$ denotes the Ripley's K-function, defined as

$$K(r) := \lambda^{-1} E\{\text{number of events within a distance } r \text{ of an arbitrary event}\}.$$

The estimator for $K(r)$ is given by

$$\hat{K}(r) = \frac{S^2}{D(D-1)} \sum_{i=1}^{D} \sum_{j=1}^{D} w_{ij} I(0 < d_{ij} < r),$$

where $S^2$ denotes the area in the object space corresponding to the image being analyzed, $w_{ij}$ denotes the Ripley's isotropic edge correction weights[27], $D$ denotes the total number of localizations, and $d_{ij}$ denotes the distance between the $i$th and $j$th localizations. The indicator function is defined as

$$I(0 < d_{ij} < r) := \begin{cases} 1 & \text{if } 0 < d_{ij} < 1 \\ 0 & \text{otherwise} \end{cases}.$$

**Estimating Ripley's K-function using multiple images.** When estimating $L(r) - r$ for a particular image analysis approach from a total of $D$ localizations distributed among $M$ images,

$$\hat{L}(r) - r = \sum_{m=1}^{M} \left( \frac{D^m}{D} \right) \hat{K}_m(r) - r,$$

where $D^m$ denotes the number of localizations obtained from the $m$th image and $\hat{K}_m(r)$ denotes estimates of the Ripley's K-function calculated using the localization obtained from that image.

**Estimating pair-correlations for an image analysis approach.** Estimates of the pair-correlation results for an image analysis approach, denoted as $a$, were calculated by a weighted averaging of pair-correlation estimates from multiple simulated images as follows. A total of $M$ images containing $D$ single molecules were simulated. A set of localizations of size $D_a^m$ were obtained by applying analysis approach $a$ to the $m$th image, for $m = 1, \ldots, M$. Pair-correlations estimates $\hat{g}_a^m(r)$ were calculated independently for each set of $D_a^m$ localizations using a MATLAB

implementation of the approach in ref. [28]. The weighted-average pair-correlation estimates for each analysis approach $a$ is then calculated as

$$\hat{g}_a(r) = \sum_{m=1}^{M} \left( \frac{D_a^m}{D_a} \right) \hat{g}_a^m(r),$$

where $D_a = D_a^1 + D_a^2 + \cdots + D_a^M$. The pair-correlation results shown in Fig. 2 were calculated using $M = 2000$ images containing $D = 250,000$ molecules.

**Determining the algorithmic resolution limit**. The radius of correlation for analysis approach $a$ is defined as

$$\rho := \inf_{r>0} \{r : g_a(r') = 1 \text{ for all } r' \in [r, \infty)\},$$

when $a$ analyzes completely spatially random data. To estimate $\rho$ from $\hat{g}_a(r)$, the scheme presented in Supplementary Note 9 was implemented. For a set $\mathcal{R} = \{r_1, \ldots, r_m\}$ of finely, regularly spaced proposal values of $\rho$,

$$\hat{\rho} = *argmin_{r_i \in \mathcal{R}} T(r_i)$$

where

$$T(r_i) = (m - i + 1)^{-1} (m - i)^{-1} \sum_{j=i}^{m} (\hat{g}_a(r_j) - \bar{g}_i)^2$$

and $\bar{g}_i = (m - i + 1)^{-1} \sum_{j=i}^{m} \hat{g}_a(r_j)$. Algorithmic resolution limit $\alpha$ is then estimated as $\hat{\rho}/2$.

**Calculating resolution adjusted Ripley's $K$-function**. For the algorithmic resolution limit $\alpha$ for a specific image analysis approach determined as described above, $L_{2\alpha}(r) - r$ is calculated as

$$L_{2\alpha}(r) - r = \sqrt{\frac{K_{2\alpha}(r) + 4\pi\alpha^2}{\pi}} - r,$$

where $K_{2\alpha}(r) = K(r) - K(2\alpha)$. See Supplementary Note 9 for details regarding the determination of $2\alpha$ for each image analysis approach from the corresponding pair-correlation results.

**Analyzing experimental superresolution data**. Let $\mathcal{D}_i$ denote the $i$th dataset consisting of $M_i$ images $\{\mathcal{I}_1^i, \ldots, \mathcal{I}_{M_i}^i\}$ for $i = 1, \ldots, N$. Let $\mathcal{D}_1$ denote the set of acquired experimental images. The remaining datasets $\mathcal{D}_2, \ldots, \mathcal{D}_N$ were generated by summing images from $\mathcal{D}_1$ as follows:

$$\mathcal{I}_m^i = \sum_{j=m_0'}^{m'} \mathcal{I}_j^1,$$

where $m_0' = m(i - 1) + 1$, $m' = m \cdot i$ for $m = 1, \ldots, M_i$. Here, $M_i$ denotes the number of images in dataset $\mathcal{D}_i$ and is calculated as $M_i = \lfloor M_1/i \rfloor$. Each of these datasets were analyzed independently using the image analysis approaches described above. The experimental dataset analyzed for Fig. 4 consisted of $M_1 = 50,000$ images.

**Reconstructing superresolution images**. Superresolution images were reconstructed from a set of localizations by simulating Gaussian profiles (as described above in the simulation of clathrin-coated pits) centered at each localization. The superresolution image was generated using a pixel size of $1/63$ µm in the object space which corresponds to $1/16$th the object space pixel size of the acquired images. Each Gaussian profile was simulated using $\sigma = 17$ nm for the width parameter and 100 detected photons. The value for $\sigma$ corresponds to the limit of the localization accuracy calculated using the average number of photons detected from each localized fluorophore and the average background noise associated with each localization.

**Software**. Region of interest identification using wavelet-filtering or global-thresholding followed by fitting with either Airy or Gaussian profiles was performed using custom programs developed with the MIATool software framework[29] in Java. The ThunderSTORM[23] and SimpleFit[24] software packages were used with default settings for the various options within the software. The QuickPALM[25] software was used with the Full Width at Half Maximum = 2 setting to match the width of the single molecule or clathrin-coated pit being localized. Calculations for $L(r) - r$, pair-correlations, and $L_{2\alpha}(r) - r$ were performed using custom-developed scripts in the MATLAB programming environment (The MathWorks, Inc., Natick, MA). All figures were similarly prepared using MATLAB.

**Data and code availability**

The data that support the findings in this study are available upon reasonable request to the corresponding authors. The software related to the analysis presented in this paper is available at https://github.com/eakcohen/algorithmic-resolution and http://www.wardoberlab.com/software.

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

## Acknowledgements

The authors would like to thank Professor Niall Adams, Department of Mathematics, Imperial College London for valuable discussions and input on implementing change point methods to estimate algorithmic resolution limits. This work was supported in part by the National Institutes of Health (R01GM085575).

## Author contributions

E.A.K.C. and R.J.O. conceived the methodology presented here, developed the mathematical theory underlying the methodology, and wrote the manuscript. A.V.A., E.A.K.C., and S.R. performed the calculations and prepared the figures for the results.

## Additional information

**Competing interests:** The authors declare no competing interests.

