## [Peer Review File · Nature Communications]

Reviewers' Comments:

Reviewer #1:

Remarks to the Author:

This paper explores the performance of localization algorithms and their potential to give misleading results. It is an important issue, especially given the increasingly central role image analysis plays in microscopy. The authors take an original approach to characterising algorithm performance and have developed some new metrics which are likely to prove useful to the field. However, there are some serious limitations both in the work and in its presentation. If these can be addressed I think the paper would be suitable for publication.

Major issues

1) The paper starts out discussing localisation microscopy and resolution, but the majority of the paper does not in fact use localisation microscopy. As far as I can tell there is no localisation microscopy in the main paper at all. This is both confusing and a lost opportunity to demonstrate the importance of the issue they have raised to the field at large.

2) In figure 1, the excitation density is very high. It is therefore completely unsurprising that the algorithms are not performing well – this is well outside the regime where you would expect them to be able to localize successfully. There is a short discussion of the issue of density in the paper but it really is the absolutely central issue here and the information relating to it should not be buried in the supplementary information. We can see from Figure 1 that the algorithm fails – unsurprisingly – when the density is high. Is it able to localise successfully for a typical localisation imaging experiment?

3) Leading from this is the issue that only single molecule fitting algorithms are tested. How do multi-emitter algorithms perform? This is available already in ThunderSTORM, and is clearly more suitable for the type of data shown in Figure 1 than a single emitter fitter. I don't think that multi-emitter algorithms will be completely accurate, but I think they may perform substantially better.

4) I am unconvinced that Figure 2 is useful. As far as I can tell this is not a localization microscopy simulation. So it's totally unsurprising that separations below about 350nm cause issues – this has been established before as an issue for localization algorithms (see references in point 5). So all this shows is that algorithms perform in the expected way. It would be much more interesting to look at the effect of density on performance, or to have some localization microscopy experiments on

clathrin pits to see how the performance changes with different resolution techniques – this would actually make the paper relevant and important in terms of super-resolution, and might turn up some unexpected results.

5) I would like to see experimental confirmation of the predicted tubulin results – this should not be difficult as tubulin is the standard sample for localization microscopy. This could be done by taking a very low density dataset, and aggregating frames to check the proportion of inaccurate localisations changes in the expected way.

6) The paper is very sparsely referenced. There has been a substantial number of previous papers on the effect, in localisation microscopy, of fluorophores being so close together that they cannot be accurately localised and how this is a limitation e.g. Wolter Opt Ex 2011, Small Biophys J 2009, Van de Linde, J Biotech 2010, Fox-Roberts, Nat Comms, 2017, and also efforts to make simulations to help avoid this effect e.g. Sinko Biomed Opt Express 2014.

Minor issues

1) The introductory paragraph misses out the important point that localization microscopy images (and the localized positions of features etc) are data derived from images, rather than images, and therefore 'resolution', in so far that it exists, is variable and not very well defined.

2) There was not much discussion of the stochastic nature of the simulations/experimental distribution, which is at the heart of the issue they describe. Even if, on average, features are a micron apart, you will get a substantial proportion of molecules closer than the 300-400nm lengthscale at which standard single molecule fitting techniques run into problems.

3) The terms 'lowest' and 'highest' resolution are used in a rather confusing way, as generally 'high' resolution refers to values of resolution which are actually low.

4) I think, for the equation at the bottom of page 3, it should be made clear immediately that alpha will be strongly algorithm dependent – as it is initially presented this does not come across.

Reviewer #2:

Remarks to the Author:

Review of “Resolution limit of image analysis algorithms” by Cohen, Abraham, and Ober.

This paper reports experiments related to single molecule localization microscopy (SMLM). The authors use an analysis of Ripley’s functions to define a new “algorithmic resolution.” This quantity indicates the resolution at which SMLM algorithms start to give poor results, as indicated by non-ideal behavior of Ripley’s function. I think the paper is mostly well done and potentially useful, but I feel there are a few important things the authors have overlooked which leave me with mixed feelings about the paper. My comments are mainly in relation to the main text, not the lengthy supplement.

1. The authors begin by imaging clathrin coated pits in a cell membrane using immunocytochemistry and widefield epifluorescence microscopy. In this situation, each clathrin coated pit will be labeled by several fluorescent dye molecules. They then analyze the acquired image using SMLM software such as ThunderSTORM or QuickPALM. Next they analyze the resulting super-resolution data using Ripley’s L function, which indicates apparent clusters of clathrin coated pits (Fig 1d). This is already problematical because the authors are not performing single molecule localization microscopy (for example PALM or dSTORM protocols). In SMLM experiments (and in SMLM analysis) we assume that the molecules are randomly photoactivated at a low density such that the imaged point spread functions do not overlap. Acquiring a long sequence of images and reconstruction of the determined molecular coordinates leads to a super-resolution image. Thus it seems to me that the authors are trying to use SMLM software for something it was not intended to do and it is not a surprise that it does not work properly. On the other hand, this is an interesting experiment and it is important to analyze such data. SMLM software might be suitable, but at much lower densities than what are present in Figure 1a. Also in this situation TIRF microscopy would be preferred but I understand that it is not always available.
2. The authors introduce algorithmic resolution, but it seems to me that what they have actually done is to rediscover the “crowded field problem” in SMLM. Here there is a rich literature which the authors do not consider. Examples include [1–7] and many more. In other words, it seems to me that the proposed algorithmic resolution is a measure of when the density of molecules in an image becomes a big enough problem as to create anomalous behavior in Ripley’s L function.
3. Moving on, the authors present the simulation shown in Figure 1e. The whole paper seems to hinge on the observation that Ripley’s L- function does not reveal a random distribution of molecules when analyzing the image in Figure 1e (reproduced here by copy-pasting from the PDF).

Figure 1e

The authors state that the data in figure 1e is a “Simulated image of clathrin-coated pits located at completely spatially random locations.” However after running SMLM software such as ThunderSTORM or QuickPALM on this image, Ripley’s L function indicates the existence of clusters of molecules (Fig 1h). The problem I have is that clusters are very clearly visible in this image, and the image does not appear at all random. Somehow the message I got reading the main text is that the whole paper is based on this single observation, which I believe is an erroneous one because I can clearly see clusters in this image. It is not a surprise that Ripley’s L function also finds them. Again I think the density is too high for the standard SMLM algorithms (those not specially designed for high density data as mentioned above) such as those the authors used (but this image is closer to that limit than Fig 1a). Furthermore, running SMLM software on this image does not change the positions of the simulated molecules, and so there should not be differences between the different approaches. Why this happens is unclear but it again has to do with the density problem.

4. The authors should consider the supplementary information of “Local dimensionality determines imaging speed in localization microscopy” [8] which also explores the issue of when the density of molecules becomes too high, but does so in 1 dimension, compared to the more complete 2D analysis in Fig.2. What about the multi-emitter fitting method present in ThunderSTORM (which was used in [8])?
5. The presented analysis using Ripley’s functions is incomplete without reports of the F1 score or Jaccard index, RMS error, and so on, as is done in [9]. Somehow it seems like the authors are conflating whether SMLM software can correctly find all the molecules with the presence of clusters, which may or may not be artificial.
6. The title of the paper seems to indicate that the paper is about the (measuring the) resolution of the super resolution image that can be obtained. This is not the case at all and is a different topic. I think the paper title should be “Evaluation of the effect of molecular density on conventional SMLM algorithms” or something.

7. ThunderSTORM's default setting uses wavelet-based pre filtering followed by Gaussian fitting, so this is similar to the author's "wavelet" method, and it is not surprising that the results are similar.

I think my issues are mainly coming from the way the paper is written and not the actual findings. I think that if the authors reorganize the paper and rewrite the main text my opinions about it would be improved.

1. J. Min, S. J. Holden, L. Carlini, M. Unser, S. Manley, and J. C. Ye, "3D high-density localization microscopy using hybrid astigmatic/ biplane imaging and sparse image reconstruction," *Biomed. Opt. Express* **5**, 3935 (2014).
2. S. J. Holden, S. Uphoff, and A. N. Kapanidis, "DAOSTORM : an algorithm for high density super-resolution microscopy," *Nat. Methods* **8**, 279–280 (2011).
3. J. Min, C. Vonesch, H. Kirshner, L. Carlini, N. Olivier, S. Holden, S. Manley, J. C. Ye, and M. Unser, "FALCON: fast and unbiased reconstruction of high-density super-resolution microscopy data.," *Sci. Rep.* **4**, 4577 (2014).
4. M. Ovesný, P. Křížek, Z. Švindrych, and G. M. Hagen, "High density 3D localization microscopy using sparse support recovery.," *Opt. Express* **22**, 31263–76 (2014).
5. H. P. Babcock, J. R. Moffitt, Y. Cao, and X. Zhuang, "Fast compressed sensing analysis for super-resolution imaging using L1-homotopy," *Opt. Express* **21**, 28583 (2013).
6. H. Babcock, Y. M. Sigal, and X. Zhuang, "A high-density 3D localization algorithm for stochastic optical reconstruction microscopy," *Opt. Nanoscopy* **1**, 1 (2012).
7. F. Huang, S. L. Schwartz, J. M. Byars, and K. A. Lidke, "Simultaneous multiple-emitter fitting for single molecule super-resolution imaging," *Biomed. Opt. Express* **2**, 1377–93 (2011).
8. P. Fox-Roberts, R. Marsh, K. Pfisterer, A. Jayo, M. Parsons, and S. Cox, "Local dimensionality determines imaging speed in localization microscopy," *Nat. Commun.* **8**, 13558 (2017).
9. D. Sage, H. Kirshner, T. Pengo, N. Stuurman, J. Min, S. Manley, and M. Unser, "Quantitative evaluation of software packages for single-molecule localization microscopy," *Nat. Methods* **12**, 717–724 (2015).

Reviewer #3:

Remarks to the Author:

The authors investigate the effect of algorithmic limitations on the ability to analyse spatial structure in fluorescence images, primarily by investigating Ripley's K correlations in clustered vs simulated CSR data.

I really enjoyed reading this manuscript – it's one of those elegantly simple observations that you're kind of surprised no one has identified before.

Overall, I support publication of the work, once the following two main criticisms are addressed:

- As far as I can see there is no discussion of the effect on image SNR on algorithmic performance. For a given algorithm, surely dropping the SNR by an order of magnitude will increase the observed algorithmic resolution limit α . Indeed, I think I dimly remember reading something to this effect in Ober's 2006 PNAS paper – the "real" Rayleigh criterion for two adjacent spots is actually a function of image SNR. Surely then α is not just a constant, but rather is some continuous function of SNR for a given algorithm? If so, this is a pretty critical point which needs to be discussed.
- This work is exclusively focused on Ripley's K/ pair correlation analysis (although I would guess it generalizes to other spatial point process statistics). However, an equally important analysis tool for analysis of object organization is cluster analysis. The authors should how discuss this work relates to cluster analysis (eg analysis of number of particles per cluster) – I mean in the discussion not really in terms of extra analysis, or at least identify it as an important point for further investigation.

Minor points

- If there is underlying structure in your image, eg as for the microtubules dataset discussed, is it going to affect the $L(r)$ - r analysis, the algorithm limit and any correction factors, as per the 2017 Fox-Roberts et al Nat Comms paper?
- On page 2 you could do with a bit more lead in on how to interpret $L(r)$ - r , eg ok so CSR is flat, but how does correlation/ anticorrelation at a given distance scale show up on an $L(r)$ - r plot.
- Page 3. I read Fig 1l as Fig 11 and got lost. Maybe Fig 1L?
- Page 4: "However, knowing the algorithmic resolution limit a of an algorithm allows us to define a resolution-corrected Ripley's K_{2a} -function and resolution-corrected $L_{2a}(r)$ - r for $r \geq 2a$ " –

This is really neat. It would be very nice to mention how calculating the resolution corrected Ripley's K etc should be approached for localization microscopy

- It would be fun (but completely non-essential) to test what α value a multi-emitter fitting algorithm gave on the test CSR data in comparison to the single emitter fitting algorithms tested here.

Response to reviewers' comments:

We want to thank the reviewers for their positive comments and their detailed suggestions that have helped to improve the manuscript. We made extensive changes to the manuscript, by adding additional data and by rewriting and newly writing several sections to address the comments by the reviewers.

For example, we have now added significant material to investigate how algorithm resolution affect the analysis in dependence of the different object density levels. We have also added significant material related to localisation based super resolution microscopy.

Further, we have taken into account comments of some of the reviewers that we thought we best address by anonymising the algorithms that we investigate. We did our best to use the software packages, using default parameter settings. However, we cannot exclude that a more expert user might obtain different results. We therefore, while citing the papers that gave rise to the various software packages, only refer to them now as Algorithm 1, Algorithm 2 etc. At the end of this response we do, however, identify the algorithms so that the reviewers have access to the information.

Reviewer #1:

1) The paper starts out discussing localisation microscopy and resolution, but the majority of the paper does not in fact use localisation microscopy. As far as I can tell there is no localisation microscopy in the main paper at all. This is both confusing and a lost opportunity to demonstrate the importance of the issue they have raised to the field at large.

We have substantially restructured the manuscript and added further material to put much more emphasis on localisation based single molecule microscopy. For example, we had put a significant amount of the material on localization based superresolution in the supplementary material, which has now been put into the main manuscript. This is in addition to the new material that was added in response to the other questions related to localisation based single molecule microscopy.

2) In figure 1, the excitation density is very high. It is therefore completely unsurprising that the algorithms are not performing well – this is well outside the regime where you would expect them to be able to localize successfully. There is a short discussion of the issue of density in the paper but it really is the absolutely central issue here and the information relating to it should not be buried in the supplementary information. We can see from Figure 1 that the algorithm fails – unsurprisingly – when the density is high. Is it able to localise successfully for a typical localisation imaging experiment?

The reviewer brings up a very important issue, the role of density of the objects in the analysis of object based data. We completely agree that the data in Figure 1 is challenging (in fact it is not a superresolution image, but a standard microscopy image of clathrin coated pits. Such images have been quite frequently analysed in the cell biology literature.) We used it to motivate the actual study, as in fact, such data has been wrongly

analysed in other publications by overlooking the resolving capability of the algorithm that is being used. In fact what motivated our current study was a publication that performed spatial statistics on clathrin-coated pit data using an algorithm that is also standardly used in single molecule data analysis and is one of the algorithms analysed in this paper. This can be justified because the coated pits can be well described by 2D Gaussian profiles with variance parameters close to those typically used in single molecule microscopy. This is not surprising considering that the size of a clathrin coated pit is usually measured to be less than 100 nm in diameter.

While we very much agree that high density is an issue, the main point of this paper is that different algorithms can perform very differently, due to the algorithmic resolution limit that is characteristic of each algorithm. Even algorithms that claim to be able to do well in the presence of higher density data exhibit significant problems. We demonstrate in the Supplementary Material that this limit is independent of the density of the objects (up to a certain point at which the algorithm breaks down). We here propose a methodology to identify this limit and how to analyse data, in the presence of these fundamental limitations.

The interplay of the algorithmic resolution limit and object density is an important aspect of this paper. Through reorganization of the manuscript (bringing material from the supplementary material to the main text and adding new materials) we hope to have made this much clearer now. We would like to specifically draw attention to the notion of “probabilistic resolution” that we introduce here and that clarifies the relationship between algorithmic resolution and object density. We also use this notion to carefully and quantitatively consider the design of localisation based superresolution experiments.

3) Leading from this is the issue that only single molecule fitting algorithms are tested. How do multi-emitter algorithms perform? This is available already in ThunderSTORM, and is clearly more suitable for the type of data shown in Figure 1 than a single emitter fitter. I don't think that multi-emitter algorithms will be completely accurate, but I think they may perform substantially better.

We agree with the reviewer's comment and have now included an analysis and discussion of multi-emitter algorithms. In fact, we had analysed the multi-emitter algorithm of ThunderSTORM before. But we had decided not to include it due the non-ideal behaviour of the algorithm. However, on further reflection, we have decided to include this analysis, as we expect that other readers would want to see such an analysis. At least regarding algorithmic resolution this multi-emitter algorithm performs worse than the single emitter algorithm. This appears to be due to errors introduced for scenarios that are handled properly by the single emitter version.

4) I am unconvinced that Figure 2 is useful. As far as I can tell this is not a localization microscopy simulation. So it's totally unsurprising that separations below about 350nm cause issues – this has been established before as an issue for localization algorithms (see references in point 5). So all this shows is that algorithms perform in the expected way. It would be much more interesting to look at the effect of density on performance, or to have some localization microscopy experiments on clathrin pits to see how the performance

changes with different resolution techniques – this would actually make the paper relevant and important in terms of super-resolution, and might turn up some unexpected results.

Spatial statistics and object based analysis plays a very important role not only in localization based superresolution microscopy but also in “conventional” microscopy experiments where cellular structures are analyzed. In these studies, in contrast to single molecule microscopy, the density cannot be controlled. It is therefore very important that the experimentalist can understand the limitations of the analysis. Unfortunately, this has not been case in the past, and in fact highly misleading studies have been published. A major objective of this manuscript is to help to develop methods that do allow data to be analyzed without the deficiencies of the prior approaches.

This part of the paper is primarily addressed to the community of cell biologists who study these and related question using “conventional” microscopy techniques. However, as some members of the single molecule community are also advocating high density approaches (often in tracking applications) these studies should also be of interest to single molecule microscopists.

We have also added an extensive analysis of how changing the density of localisation experiments impacts the probability that an individual single molecule can be resolved.

5) I would like to see experimental confirmation of the predicted tubulin results – this should not be difficult as tubulin is the standard sample for localization microscopy. This could be done by taking a very low density dataset, and aggregating frames to check the proportion of inaccurate localisations changes in the expected way.

To address the reviewer’s suggestion we have now added an analysis of a tubulin dataset (from a single molecule challenge) to investigate the role of density and algorithmic resolution. We used data from the single molecule challenge as having access to the ground truth allowed us to check the proportion of accurate localizations. We have verified that the theoretical probabilistic resolution very closely matches the proportion of accurate localizations across a range of density values.

6) The paper is very sparsely referenced. There has been a substantial number of previous papers on the effect, in localisation microscopy, of fluorophores being so close together that they cannot be accurately localised and how this is a limitation e.g. Wolter Opt Ex 2011, Small Biophys J 2009, Van de Linde, J Biotech 2010, Fox-Roberts, Nat Comms, 2017, and also efforts to make simulations to help avoid this effect e.g. Sinko Biomed Opt Express 2014.

We thank the review for the suggested references, many of which have now been added to our list of references. (We had made a mistake believing that there was a more strict limit on the number of citations than is the case.)

Minor issues

1) The introductory paragraph misses out the important point that localization microscopy images (and the localized positions of features etc) are data derived from images, rather

than images, and therefore 'resolution', in so far that it exists, is variable and not very well defined.

We hope that the changes made to the manuscript, including much greater emphasis on localization microscopy address this point by the reviewer.

2) There was not much discussion of the stochastic nature of the simulations/experimental distribution, which is at the heart of the issue they describe. Even if, on average, features are a micron apart, you will get a substantial proportion of molecules closer than the 300-400nm lengthscale at which standard single molecule fitting techniques run into problems.

We very much agree with the reviewer on this point. This is very much the motivation that underlies the approach for the determination of the algorithmic resolution limit. The data that leads to the estimation of the pair correlation function, is completely spatially random. This means that statistically all possible configurations are explored, from large to small distances, including different multi-/single-emitter configurations.

Importantly, the reviewer's comment supports one further aspect of our approach, in that even for "low-density" samples resolution problems arise, although with lower probability as analysed through our concept of probabilistic resolution which we have now moved from the Supplementary Material into the main manuscript.

3) The terms 'lowest' and 'highest' resolution are used in a rather confusing way, as generally 'high' resolution refers to values of resolution which are actually low.

We very much appreciate the reviewer's comment as this has prompted us to revise the manuscript so that we are now use 'smallest' and 'largest' when referring to the algorithmic resolution limit.

4) I think, for the equation at the bottom of page 3, it should be made clear immediately that alpha will be strongly algorithm dependent – as it is initially presented this does not come across.

We thank the reviewer for making this important point. We have clarified in the paper that an algorithm resolution limit is specific to an individual algorithm.

Reviewer #2:

1. *The authors begin by imaging clathrin coated pits in a cell membrane using immunocytochemistry and widefield epifluorescence microscopy. In this situation, each clathrin coated pit will be labeled by several fluorescent dye molecules. They then analyze the acquired image using SMLM software such as ThunderSTORM or QuickPALM. Next they analyze the resulting super-resolution data using Ripley's L function, which indicates apparent clusters of clathrin coated pits (Fig 1d). This is already problematical because the authors are not performing single molecule localization microscopy (for example PALM or dSTORM protocols). In SMLM experiments (and in SMLM analysis) we assume that the molecules are randomly photoactivated at a low density such that the imaged point spread functions do not overlap. Acquiring a long sequence of images and reconstruction of the determined molecular coordinates leads to a super-resolution image. Thus it seems to me that the authors are trying to use SMLM software for something it was not intended to do and it is not a surprise that it does not work properly. On the other hand, this is an interesting experiment and it is important to analyze such data. SMLM software might be suitable, but at much lower densities than what are present in Figure 1a. Also in this situation TIRF microscopy would be preferred but I understand that it is not always available.*

Thank you for making these comments as they provided the opportunity to clarify an important point in the manuscript. In fact, clathrin coated pit data has been in past analysed with single molecule localization analysis software (Legache et al, 2013⁶). While we agree that at first site this might suggest that a tool is applied to a problem for which it was not designed, we do in fact agree with the authors of (OM) that it is an appropriate choice as the coated pits can be well described by 2D Gaussian profiles and in fact the variance parameters that need to be used are close to those typically used in single molecule microscopy. This is not surprising considering that the size of a clathrin coated pit is usually measured to be less than 100 nm in diameter (see also EM studies etc.). In fact some of the very successful single molecule detection techniques had originally been developed for the detection of spot like structured imaged with conventional microscopy techniques, such as clathrin coated pits, small vesicles etc.

We have extensive experience imaging clathrin coated pits using TIRF and using conventional microscopy. For the purposes of the analysis presented here using TIRF would, in our experience, not produce significantly different images.

2. *The authors introduce algorithmic resolution, but it seems to me that what they have actually done is to rediscover the "crowded field problem" in SMLM. Here there is a rich literature which the authors do not consider. Examples include [1–7] and many more. In other words, it seems to me that the proposed algorithmic resolution is a measure of when the density of molecules in an image becomes a big enough problem as to create anomalous behavior in Ripley's L function.*

Algorithmic resolution is independent of density, both based on the theoretical consideration, but also verified (Supplementary Materials Figure 7). Having said this, the problematic effects of algorithmic resolution are more pronounced in the high density situation as the probability of two objects being within α each other is higher than for low density samples. In fact, this is one of the main points of analysis of the paper. The notion of probabilistic resolution that we introduce, and now discuss much more extensively, describes with which probability an object's detection is not affected by resolution. The probabilistic resolution depends on both the algorithmic resolution of the algorithm employed and the density of the sample. In fact, we show calculations how, in a localisation based superresolution experiment, the choice of density per frame, directly impacts probabilistic resolution. Importantly, even for very low density samples, algorithmic resolution has an effect that is, naturally, smaller than that for high density samples.

3. Moving on, the authors present the simulation shown in Figure 1e. The whole paper seems to hinge on the observation that Ripley's L-function does not reveal a random distribution of molecules when analyzing the image in Figure 1e (reproduced here by copy-pasting from the PDF). The authors state that the data in figure 1e is a "Simulated image of clathrin-coated pits located at completely spatially random locations." However after running SMLM software such as ThunderSTORM or QuickPALM on this image, Ripley's L function indicates the existence of clusters of molecules (Fig 1h). The problem I have is that clusters are very clearly visible in this image, and the image does not appear at all random. Somehow the message I got reading the main text is that the whole paper is based on this single observation, which I believe is an erroneous one because I can clearly see clusters in this image. It is not a surprise that Ripley's L function also finds them. Again I think the density is too high for the standard SMLM algorithms (those not specially designed for high density data as mentioned above) such as those the authors used (but this image is closer to that limit than Fig 1a). Furthermore, running SMLM software on this image does not change the positions of the simulated molecules, and so there should not be differences between the different approaches. Why this happens is unclear but it again has to do with the density problem.

We should mention that the appearance of completely spatially random data in fact often leads to what might be considered to be "clusters". This is a phenomena not dissimilar from being able to obtain several "6" in a row by casting dice. This would not be an indication of the absence of a uniform distribution of the outcomes 1-6. In fact, the data shown in Figures 1e and 1i is completely spatially random as per the definition (a homogeneous Poisson process). A particular manifestation of such completely spatially random processes is that locations can (randomly) be arbitrarily close and visually appear as clusters. This is what the reviewer has correctly pointed out. It is this property that has lead us to use such data as templates to analyse algorithms. It is, however, that different algorithms have different capabilities to deal with such location configurations that our analysis reveals.

4. The authors should consider the supplementary information of "Local dimensionality

determines imaging speed in localization microscopy” [8] which also explores the issue of when the density of molecules becomes too high, but does so in 1 dimension, compared to the more complete 2D analysis in Fig.2. What about the multi-emitter fitting method present in ThunderSTORM (which was used in [8])?

We have now also included the multi-emitter option of ThunderSTORM and an analysis was presented for this (please see response to Reviewer 1, comment 3 for more details). Fox Roberts et al (2017) is now cited in the paper¹⁸.

5. The presented analysis using Ripley’s functions is incomplete without reports of the F1 score or Jaccard index, RMS error, and so on, as is done in [9]. Somehow it seems like the authors are conflating whether SMLM software can correctly find all the molecules with the presence of clusters, which may or may not be artificial.

The scoring approaches that the reviewer mentions are certainly very important. The algorithmic resolution limit that is introduced and analysed here is novel and reveals different aspects of an algorithm than the scores that the reviewer mentions. We have added comments in the paper to address this issue that is certainly important for the reader to be informed about.

6. The title of the paper seems to indicate that the paper is about the (measuring the) resolution of the super resolution image that can be obtained. This is not the case at all and is a different topic. I think the paper title should be “Evaluation of the effect of molecular density on conventional SMLM algorithms” or something.

As the novel insights regarding algorithmic resolution are the focus of this paper we would like to keep this aspect of the title. This should hopefully not mislead the reader to assume that the sample density is also of relevance. But the relevance does depend in a major way on the algorithmic resolution as studied in the manuscript, both from a theoretical point of view and using simulations etc.

7. ThunderSTORM’s default setting uses wavelet-based pre filtering followed by Gaussian fitting, so this is similar to the author’s “wavelet” method, and it is not surprising that the results are similar.

This is a very valid point, the methods are very similar and therefore the similar results are to be expected.

I think my issues are mainly coming from the way the paper is written and not the actual findings. I think that if the authors reorganize the paper and rewrite the main text my opinions about it would be improved.

Based on this reviewer’s and the other reviewers’ comments we have significantly reworked the presentation of the manuscript. We hope that as a result the paper and its contributions are more clearly presented.

Reviewer #3

The authors investigate the effect of algorithmic limitations on the ability to analyse spatial structure in fluorescence images, primarily by investigating Ripley's K correlations in clustered vs simulated CSR data.

I really enjoyed reading this manuscript – it's one of those elegantly simple observations that you're kind of surprised no one has identified before.

Overall, I support publication of the work, once the following two main criticisms are addressed:

- As far as I can see there is no discussion of the effect on image SNR on algorithmic performance. For a given algorithm, surely dropping the SNR by an order of magnitude will increase the observed algorithmic resolution limit α . Indeed, I think I dimly remember reading something to this effect in Ober's 2006 PNAS paper – the "real" Rayleigh criterion for two adjacent spots is actually a function of image SNR. Surely then α is not just a constant, but rather is some continuous function of SNR for a given algorithm? If so, this is a pretty critical point which needs to be discussed.*

We thank the reviewer for this very insightful comment. We have now added additional data in Supplementary Materials 6 and comments in the main paper to address this point in detail. Yes, the 2006 results show very clearly that the standard deviation with which the distance between two points can be estimated depends on the signal levels (due to the sample, but also background, readout noise etc.). We analysed the dependence of the algorithmic resolution on the signal level. We found that for levels that are typical for single molecule experiments there is no appreciable dependence on the signal level. Only for extremely low levels did this dependence set in. This is consistent with the 2006 results as the algorithmic resolution limit for the algorithms that are investigated are of the order of Rayleigh's criterion. But for those distances we found in the 2006 paper that the dependence on the signal level is less pronounced than for extremely low levels.

- This work is exclusively focused on Ripley's K/ pair correlation analysis (although I would guess it generalizes to other spatial point process statistics). However, an equally important analysis tool for analysis of object organization is cluster analysis. The authors should how discuss this work relates to cluster analysis (eg analysis of number of particles per cluster) – I mean in the discussion not really in terms of extra analysis, or at least identify it as an important point for further investigation.*

This is again an interesting point. Typically, algorithmic resolution leads to uncertainty in the number of localizations, this might be an underestimation or an overestimation depending on the data and the algorithm used. Proposition 4 in Supplementary Material 4.1 captures this effect, demonstrating that this uncertainty disappears as the density tends to zero. It is this proposition that captures the very essence of what SMLM techniques such as PALM and STORM are trying to achieve. A direct consequence of this is that output from clustering algorithms (e.g. Rubin-Delanchy et al

2016) that act on the final set of localizations need to be treated with caution. We have now introduced into the main manuscript a thorough analysis of probabilistic resolution (the probability that an object is unaffected by resolution) for Thomas processes – a common cluster process model. Understanding the probabilistic resolution informs on how confident one could be in the deploying clustering algorithms to experimental data. This has also been included in the discussion.

Minor points

- If there is underlying structure in your image, eg as for the microtubules dataset discussed, is it going to affect the $L(r)$ - r analysis, the algorithm limit and any correction factors, as per the 2017 Fox-Roberts et al Nat Comms paper?

The algorithmic resolution limit is invariant to the type of point pattern that is being imaged – although it is defined through the analysis of CSR data. You are correct, however, that the underlying structure of the data is important as the probability that objects are affected by this resolution issue is dependent on it. This was given significant treatment in the Supplementary Material, however following also the other reviewer's comment we have now included an extensive analysis of tubulin data set and clustered data in the main text. Where the difference comes in related to CSR data is through the nearest neighbour distribution G of the point pattern. We have characterised the effect of the underlying structure of the data through the 'probabilistic resolution'.

- On page 2 you could do with a bit more lead in on how to interpret $L(r)$ - r , eg ok so CSR is flat, but how does correlation/ anticorrelation at a given distance scale show up on an $L(r)$ - r plot.

We have added additional comments to explain this property.

- Page 3. I read Fig 1l as Fig 11 and got lost. Maybe Fig 1L?

We thank the reviewer for these comments and have now addressed them by calling it Fig 1L

- Page 4: "However, knowing the algorithmic resolution limit a of an algorithm allows us to define a resolution-corrected Ripley's K_{2a} -function and resolution-corrected $L_{2a}(r)$ - r for $r \geq 2a$ " – This is really neat. It would be very nice to mention how calculating the resolution corrected Ripley's K etc should be approached for localization microscopy

This is an interesting methodological question. We have now proposed a Ripley's K -function estimator in Supplementary Materials 8.1 that can be used in a localization microscopy setting. The traditional estimator for the Ripley's K -function requires moving through every event (localization) and counting the number of other events within a distance r of it. In the situation where we split the objects across several frames we can make use of this to negate the effects of algorithmic resolution. In the presented estimator,

we now move through each point counting the number objects *that appear in a different frame only*. With a readjustment to the normalizing factor, we can now form an estimator for $K(r)$ for all $r>0$, even at distances less than the algorithmic resolution limit.

- It would be fun (but completely non-essential) to test what alpha value a multi-emitter fitting algorithm gave on the test CSR data in comparison to the single emitter fitting algorithms tested here.

As also suggested by Reviewer 2 we have added such an analysis.

Algorithm Key

Algorithm 1: Wavelet-based

Algorithm 2: ThunderSTORM (single-emitter)

Algorithm 3: SimpleFit

Algorithm 4: Global Threshold

Algorithm 5: QuickPALM

Algorithm 6: ThunderSTORM (multi-emitter)

Reviewers' comments:

Reviewer #1 (Remarks to the Author):

I have read through the revised version of the paper. I remain impressed by the insightful ideas and interesting approach the authors have taken, and I think it should ultimately be published. However, I think that there are still issues, particularly with the presentation. The authors state in their response that one part of the paper is:

'addressed to the community of cell biologists who study these and related question using "conventional" microscopy techniques. However, as some members of the single molecule community are also advocating high density approaches (often in tracking applications) these studies should also be of interest to single molecule microscopists.'

The work does indeed have implications for the work of cell biologists, but I do not think in its current form that the paper is accessible to them (I found it challenging to read even though it is roughly my area, I am very interested in it, and I had read it once before). I think it would be worth the authors simplifying their presentation further and having non-mathematical descriptions of concepts, the advantages of this approach, and its limitations.

For example, I would list the main points of the paper as:

- 1) When imaging using fluorescence microscopy points need to be much further apart than the PSF width to be localized accurately.
- 2) This can affect results both when you try to fit positions in structures where there are a number of fluorophores, and in localization microscopy.
- 3) Using simulations, you can calculate the probability that an object fitting will be affected by a misfit, or that a localization event will be misfitted.
- 4) This relies on having large amounts of data.

Which leaves me with the questions: how much data is needed in practice to evaluate an algorithm, and will this ever be a practical method of evaluation using real data rather than simulations for localization microscopy?

This is part of the reason I requested comparison to an tubulin experimental dataset with low and high density, but the authors have actually used a simulated dataset. I realize that experimental data

is more of a pain to deal with, has to be acquired, and is challenging because ground truth is not available. But that is exactly why I wanted the authors to look at it. Despite the advances in simulation over the last decade, simulated data and experimental data are not the same and differences between the two can have important implications for the performance of an algorithm. I would strongly suggest the authors use experimental data, or if not at least have a discussion in the paper of the fact that experimental data for localization microscopy will have other factors which will impact the performance (to give just one example, the activation density will not be uniform over time).

Also, I don't think the anonymization of the algorithms is helpful. It is important for the reader to see which algorithm is which. Why cut all the information which might actually be useful to people in the field out of the paper?

Minor issues:

There is a discussion at the top of page 3 about spatially random data. I don't think it conveys the difference between a distribution where points/fluorophores are uniformly distributed (i.e. equally spaced) and one where they are randomly distributed. In fact there's an unhelpful use of the term 'uniform distribution', which can be used to denote a distribution which is uniformly spaced, rather than one with a uniform probability of appearance.

References to work in the Supplementary section are not necessarily very helpful – for example, following the reference to probabilistic resolution to look at the exact definition led to Supplementary Material 3, but I then had to scroll through to 3.3 to get to the definition. It would be useful to make the references more exact. Also the issues of the probability of a particular object being affected by the resolution, and the probability of a point process of an event being affected by the resolution should be clarified i.e. the second one depends on the activation.

Reviewer #2 (Remarks to the Author):

The paper has been improved and I think the authors have done a fair job in addressing the reviewers concerns.

Reviewer #3 (Remarks to the Author):

The authors have addressed my concerns.

I have a few minor points:

- The algorithms 1-6 are not clearly identified in the methods, eg 2,3,5 are grouped together followed by jumbled up references, and the algorithms are not clearly named (where this is possible). Algorithm 6 is not clearly identified (I think I only found that it was the ThunderSTORM multi-emitter algorithm from the rebuttal. Please make an SI table listing each algorithm number, the software name, reference, any critical software settings used when running it.

- Please either zoom out (in y axis) or provide an additional figure of Fig S3a $g(r)$ vs r such that all the data are visible.

- It would be worth adding a caveat that the strange multi-emitter $g(r)$ behaviour observed for algorithm 6 may be an artefact of that particular software rather than the entire conceptual approach - more multi-emitter algorithms would need to be tested to resolve this but I think that is beyond the scope of the manuscript.

Response to reviewers

Reviewer #1 (Remarks to the Author):

I have read through the revised version of the paper. I remain impressed by the insightful ideas and interesting approach the authors have taken, and I think it should ultimately be published. However, I think that there are still issues, particularly with the presentation. The authors state in their response that one part of the paper is:

'addressed to the community of cell biologists who study these and related question using "conventional" microscopy techniques. However, as some members of the single molecule community are also advocating high density approaches (often in tracking applications) these studies should also be of interest to single molecule microscopists.'

The work does indeed have implications for the work of cell biologists, but I do not think in its current form that the paper is accessible to them (I found it challenging to read even though it is roughly my area, I am very interested in it, and I had read it once before). I think it would be worth the authors simplifying their presentation further and having non-mathematical descriptions of concepts, the advantages of this approach, and its limitations.

We thank the reviewer for the comments regarding readability by non-experts in the field of spatial statistics. We are also very much interested in the paper being as accessible as possible. Our approach was therefore to relegate all the detailed mathematical statements and derivations to the Supplementary Material, leaving the main manuscript to be a presentation of the basic ideas and results. However, we now also realize that some of the descriptions in the main manuscript could benefit from further expansion and explanation. We have therefore significantly expanded the text, including the addition of illustrative figures (added to the supplementary material). We now hope that the material is more easily accessible. As the main content of the paper is the introduction of several completely new concepts the manuscript may by necessity require more detailed reading than a more results oriented paper that relies on established concepts.

All changes are marked in the manuscript.

For example, I would list the main points of the paper as:

- 1) When imaging using fluorescence microscopy points need to be much further apart than the PSF width to be localized accurately.*
- 2) This can affect results both when you try to fit positions in structures where there are a number of fluorophores, and in localization microscopy.*
- 3) Using simulations, you can calculate the probability that an object fitting will be affected by a misfit, or that a localization event will be misfitted.*
- 4) This relies on having large amounts of data.*

Which leaves me with the questions: how much data is needed in practice to evaluate an algorithm, and will this ever be a practical method of evaluation using real data rather than simulations for localization microscopy?

Details on the methods used to evaluate the algorithm, including the amount of data, are in the Methods section. As with all estimation procedures, the quality and precision of the algorithmic resolution estimate improves with increased data. For this reason, the amount of data we use is large (250,000 molecules across 2000 images), and this produces confidence intervals (as estimated

by the bootstrapping procedure of Supplementary Material 6.2) of reasonable size. Under our theoretical framework, the algorithmic resolution is judged by its response to complete spatial randomness. This is a key point and due to the fact CSR data is spatially uncorrelated. This allows theoretical results on the effect of algorithmic resolution to be obtained (see Supplementary Material 2.3), provides a universal baseline with which to analyse performance, and contains all possible configurations of objects. Therefore, if truly CSR data can be generated in experiments it is in theory possible to estimate the algorithmic resolution. However, the quality of the estimate, and hence the size of the confidence intervals, will inevitably be linked to the amount of CSR data that can be generated. This is why we believe that simulations are the easiest, most cost effective and most accurate route to analysing an algorithm's performance.

This is part of the reason I requested comparison to an tubulin experimental dataset with low and high density, but the authors have actually used a simulated dataset. I realize that experimental data is more of a pain to deal with, has to be acquired, and is challenging because ground truth is not available. But that is exactly why I wanted the authors to look at it. Despite the advances in simulation over the last decade, simulated data and experimental data are not the same and differences between the two can have important implications for the performance of an algorithm. I would strongly suggest the authors use experimental data, or if not at least have a discussion in the paper of the fact that experimental data for localization microscopy will have other factors which will impact the performance (to give just one example, the activation density will not be uniform over time).

An experimental tubulin dataset has now been included to verify the results of the paper. The original dataset, Dataset 1, is comprised of 50000 frames. Averaging pairs of images, we are able to create Dataset 2 that consists of 25000 frames, each with double the object density. Dataset 3, formed by averaging triplets of frames, consists of 16666 frames with triple the object density. This is repeated up to Dataset 10, which consists of 5000 frames with ten times the object density. The number of localizations generated by each algorithm for each dataset are shown in Figure 3h (and Supplementary Figure 4 shows examples of the reconstructed images from these data sets). These results demonstrate two key points. The first is it can be seen that Algorithm 2, the algorithm with the smallest resolution limit, has the largest number of localizations. Furthermore, Algorithm 1, which has a similar algorithmic resolution limit, has a very similar number of localizations. However, Algorithm 3, which has an algorithmic resolution limit almost twice as large as Algorithms 1 and 2, produces far fewer localizations. This is predicted by the presented theory; a larger algorithmic resolution results in a smaller probabilistic resolution under the same molecule density, and as such we would expect fewer localizations. The second point is that the number of localizations decreases as the density increases. Again, this is predicted under our theoretical framework since we show probabilistic resolution decreases with density. This demonstrates that experimental observations on the performance of different algorithms are consistent with the findings of the paper, which have been reached from simulation methods.

Also, I don't think the anonymization of the algorithms is helpful. It is important for the reader to see which algorithm is which. Why cut all the information which might actually be useful to people in the field out of the paper?

As a matter of general principle we are always very concerned to evaluate software packages written by others without their input. While we believe we have taken great care in doing so, the use of software written by others is always very problematic as the reliance of default settings might not yield to best performance of the algorithm that an expert could potentially achieve. We

therefore thought it is better to anonymize the algorithms. This will achieve the main purpose to illustrate different behaviors of algorithms but avoids potentially publishing a suboptimal performance of a software package. In the manuscript itself we list the references in which the algorithms were published, without specifying which algorithm is which. The anonymization was also carried in response to concerns expressed by another reviewer as part of the prior revision. Should the Editor wish us to revert to explicitly listing the algorithms, we would be happy to change the manuscript accordingly.

Minor issues:

There is a discussion at the top of page 3 about spatially random data. I don't think it conveys the difference between a distribution where points/fluorophores are uniformly distributed (i.e. equally spaced) and one where they are randomly distributed. In fact there's an unhelpful use of the term 'uniform distribution', which can be used to denote a distribution which is uniformly spaced, rather than one with a uniform probability of appearance.

We use the term 'uniform distribution' under its typical definition to describe a probability distribution with constant probability density function. In the first instance in which we use it, we have now clarified that we mean the probability distribution. We have also rewritten this section and added a figure to the Supplementary Material to further clarify the concepts of complete spatial randomness, clustering and inhibition.

References to work in the Supplementary section are not necessarily very helpful – for example, following the reference to probabilistic resolution to look at the exact definition led to Supplementary Material 3, but I then had to scroll through to 3.3 to get to the definition. It would be useful to make the references more exact. Also the issues of the probability of a particular object being affected by the resolution, and the probability of a point process of an event being affected by the resolution should be clarified i.e. the second one depends on the activation.

Thank you for pointing out that some of the references to the Supplementary Material could be more precise. We have now addressed this.

Reviewer #2 (Remarks to the Author):

The paper has been improved and I think the authors have done a fair job in addressing the reviewers concerns.

We are very happy to see we have addressed the reviewer's points.

Reviewer #3 (Remarks to the Author):

The authors have addressed my concerns.

I have a few minor points:

- The algorithms 1-6 are not clearly identified in the methods, eg 2,3,5 are grouped together followed by jumbled up references, and the algorithms are not clearly named (where this is possible). Algorithm 6 is not clearly identified (I think I only found that it was the ThunderSTORM multi-emitter algorithm from the rebuttal. Please make an SI table listing each algorithm number, the software name, reference, any critical software settings used when running it.

In the revised manuscript, we elected to anonymize the algorithms. The reasoning for this is given in our response to reviewer 2, who raised a similar point. We apologize if this was not clear in our response to the reviewers. As stated above, we will be happy to change the manuscript should the Editor wish us to.

- Please either zoom out (in y axis) or provide an additional figure of Fig S3a $g(r)$ vs r such that all the data are visible.

An extra figure has now been added.

- It would be worth adding a caveat that the strange multi-emitter $g(r)$ behaviour observed for algorithm 6 may be an artefact of that particular software rather than the entire conceptual approach - more multi-emitter algorithms would need to be tested to resolve this but I think that is beyond the scope of the manuscript.

We appreciate this point and have now added a comment in the manuscript indicating that other multi-emitters might behave differently.

REVIEWERS' COMMENTS:

Reviewer #1 (Remarks to the Author):

The authors have addressed all my main concerns and I think the paper should be published.

I have one further minor suggestion: in Figure 3h, the use of 'Dataset' for the horizontal axis is not intuitively useful. I would suggest using a terms like 'Relative raw data density' or 'Relative activation probability per frame'. Then the base data would be 1, and the numbering would be the same as it is currently. There also seems to be a stray 10 in the middle of the horizontal axis on the copy of the paper I got.

I disagree with the decision to keep the algorithms anonymous. I hope the editor will provide input on this.

Response to reviewers

Reviewer #1 (Remarks to the Author):

>>The authors have addressed all my main concerns and I think the paper should be published.

We are happy to see we have now addressed your concerns. Many thanks for your carefully reading and insightful comments that have no doubt led to an improved manuscript.

>>I have one further minor suggestion: in Figure 3h, the use of 'Dataset' for the horizontal axis is not intuitively useful. I would suggest using a terms like 'Relative raw data density' or 'Relative activation probability per frame'. Then the base data would be 1, and the numbering would be the same as it is currently.

We are happy to make the change.

>>There also seems to be a stray 10 in the middle of the horizontal axis on the copy of the paper I got.

Thank you for noticing this editing error. This has now been removed.

>>I disagree with the decision to keep the algorithms anonymous. I hope the editor will provide input on this.

The editor has advised us to remove the anonymization, which we have now done.